# Using ground radar overlaps to verify the retrieval of calibration bias estimates from space-borne platforms

Irene Crisologo[1] and Maik Heistermann[1]

[1]Institute of Environmental Sciences and Geography, University of Potsdam, Potsdam, Germany

**Correspondence:** Irene Crisologo (crisologo@uni-potsdam.de)

**Abstract.** Many institutions struggle to tap the potential of their large archives of radar reflectivity: these data are often affected by miscalibration, yet the bias is typically unknown and temporally volatile. Still, relative calibration techniques can be used to correct the measurements a posteriori. For that purpose, the usage of space-borne reflectivity observations from the Tropical Rainfall Measuring Mission (TRMM) and Global Precipitation Measurement (GPM) platforms has become increasingly popular: the calibration bias of a ground radar is estimated from its average reflectivity difference to the space-borne radar (SR). Recently, Crisologo et al. (2018) introduced a formal procedure to enhance the reliability of such estimates: each match between SR and GR observations is assigned a quality index, and the calibration bias is inferred as a quality-weighted average of the differences between SR and GR. The relevance of quality was exemplified for the Subic S-band radar in the Philippines which is much affected by partial beam blockage.

The present study extends the concept of quality-weighted averaging by accounting for path-integrated attenuation (PIA), in addition to beam blockage. This extension becomes vital for radars that operate at C- or X-band. Correspondingly, the study setup includes a C-band radar which substantially overlaps with the S-band radar. Based on the extended quality-weighting approach, we retrieve, for each of the two ground radars, a time series of calibration bias estimates from suitable SR overpasses. As a result of applying these estimates to correct the ground radar observations, the consistency between the ground radars in the region of overlap increased substantially. Furthermore, we investigated if the bias estimates can be interpolated in time, so that ground radar observations can be corrected even in the absence of prompt SR overpasses. We found that a moving average approach was most suitable for that purpose, although limited by the absence of explicit records of radar maintenance operations.

## 1 Introduction

Weather radar observations are key to quantitative precipitation estimation (QPE) with a large spatial coverage and a high resolution in space and time (in the order of $10^2 - 10^3$ meters, and $10^0 - 10^1$ minutes). Yet, the indirect nature of the precipitation retrieval paves the way for a multitude of systematic estimation and measurement errors. We define *estimation errors* as errors

that occur in the retrieval of the precipitation rate R from the radar's prime observational target variable, the radar reflectivity factor Z. These errors are caused mainly by the unknown microphysical properties of the target–be it meteorological or non-meteorological. Before that, *measurement errors* affect the observation of Z through a multitude of mechanisms that can accumulate as the beam propagates through the atmosphere (such as beam blockage, or path-integrated attenuation (PIA)). In addition, the prominence of these measurement errors heavily depends on scenario-specific interaction of factors such as radar bandwidth, beam width, obstacles in the direct and wider vicinity, topography in the radar coverage, atmospheric refractivity, or the microphysical properties of precipitation along the beam's propagation path. Much has been written about these sources of uncertainty, and much has been done to address them adequately (see Villarini and Krajewski (2010) for an extensive review).

Yet, the single-most contribution of uncertainty to radar-based QPE often comes from the (mis)calibration or (in)stability of the radar instrument itself (Houze et al., 2004) which can also vary in time (Wang and Wolff, 2009). Apart from the simple fact that miscalibration can easily deteriorate the accuracy of precipitation estimates by an order of magnitude, calibration issues become particularly annoying if weather radars are operated in a network where the consistency of calibration between radars is a prerequisite for high-quality radar mosaics (see e.g. Seo et al. (2014)).

There are various options to carry out and monitor the calibration of a radar instrument in an operational context through absolute calibration techniques (based on a well-defined reference noise source, see Doviak and Zrnić (2006) for an overview). Yet, to the reflectivity that is already measured and recorded, any changes to the instrument's calibration are irrelevant. In such a case, relative calibration techniques can be used to correct the measurements a posteriori. Many institutions have archived large radar reflectivity records over the years, but they struggle to tap the potential of these data due to unknown and temporally volatile calibration biases. And while radar polarimetry offers new opportunities to address calibration issues, many archived data still originate from single-polarization radars.

As to relative calibration, the usage of rain gauge observations is typically not recommended, not only due to issues of representativeness in space and time, but also due to the fact that a comparison between *R*, as observed by rain gauges, and *R*, as retrieved from radar reflectivities, lumps over measurement *and* estimation uncertainties. As an alternative, the usage of space-borne reflectivity observations from the Tropical Rainfall Measuring Mission (TRMM) and Global Precipitation Measurement (GPM) platforms has become increasingly popular over the recent years. Measurement accuracies of both TRMM and GPM are reported to have excellent calibration (within < 1dB) (Kawanishi et al., 2000; Hou et al., 2013), and thus can be used as a reference to calibrate reflectivity. Moreover, a major benefit of relative calibration is that it allows for a posteriori correction of historical data.

In a recent study for an S-band radar in the Philippines, Crisologo et al. (2018) adopted a technique to match ground radar (GR) and space-borne radar (SR) observations. That technique was originally suggested by Bolen and Chandrasekar (2003), then further developed by Schwaller and Morris (2011), and recently by Warren et al. (2018). The underlying idea of that technique is to match observations based on the geometric intersection of SR and GR beams. That way, the algorithm confines the comparison to locations where both instruments have valid observations, and avoids artefacts from interpolation or extrapolation. In that context, Crisologo et al. (2018) demonstrated that explicitly taking into account the quality of the GR observations is vital to enhance the consistency between SR and GR reflectivity measurements, and thus to estimate the

calibration bias more reliably. The relevance of quality was exemplified by considering partial beam blockage: for each GR bin, a quality index between 0 and 1 was inferred from the beam blockage fraction. These quality indices were then used to compute a quality-weighted average of volume matched GR reflectivities.

The present study aims to extend the approach of Crisologo et al. (2018) in several respects:

1. We extend the framework to account for the quality of GR observations by introducing path integrated attenuation (PIA) as a quality variable, in addition to partial beam blockage. Instead of attempting to correct GR reflectivities for PIA, we explicitly acknowledge the uncertainty of any PIA estimate by assigning a low weight to any GR bins that are substantially affected by PIA. In order to investigate the role of PIA, we include a C-band weather radar in the present study, in addition to the S-band radar included by Crisologo et al. (2018).

2. We verify the ability to estimate the GR calibration bias from SR overpass data by evaluating the consistency of GR reflectivity measurements in a region of overlap, before and after bias correction.

3. We investigate whether estimates of GR calibration bias, as obtained from SR overpass data, can be interpolated in time in order to correct GR reflectivity observations for miscalibration, even for those times in which no suitable SR overpasses were available.

The last item—the interpolation of bias estimates in time—would be a key requirement towards actually tapping the potential of the fundamental concept in research and applications: if we aim to use SR overpass data for monitoring GR calibration bias, and for a homogeneous correction of archived GR reflectivities, we have to assume that those bias estimates are, to some extent, representative in time. Crisologo et al. (2018) found that the bias estimates for the SUB S-band radar exhibited a substantial short-term temporal variability, and stated that they *"would not expect changes in calibration bias to occur at the observed*
*frequency, amplitude, and apparent randomness."* By investigating whether such bias estimates can be interpolated in time, the present paper will investigate whether the apparently "volatile" behaviour of calibration bias is not a mere artefact of the estimation procedure, but a real property of the investigated radar systems.

Section 2 of the present paper will describe the study area and the underlying radar data sets and section 3 will outline the methodologies of matching GR and SR as well as GR and GR observations, the quantification of beam blockage and PIA,
and the quality-based framework for bias estimation. Section 4, we will show and discuss the various inter-comparison results followed by the conclusion in Section 5.

## 2   Data and Study Area

The Philippines' weather agency, known as the Philippine Atmospheric, Geophysical, and Astronomical Services Administration (PAGASA), maintains a network of 10 ground radars all over the country, of which 8 are single-polarization S-Band
radars and 2 are dual-polarization C-Band radars. Two of the longest running radars are Subic (SUB) and Tagaytay (TAG). Between the two radars lies Manila Bay, bordered on the east by Metro Manila, the country's most densely populated area

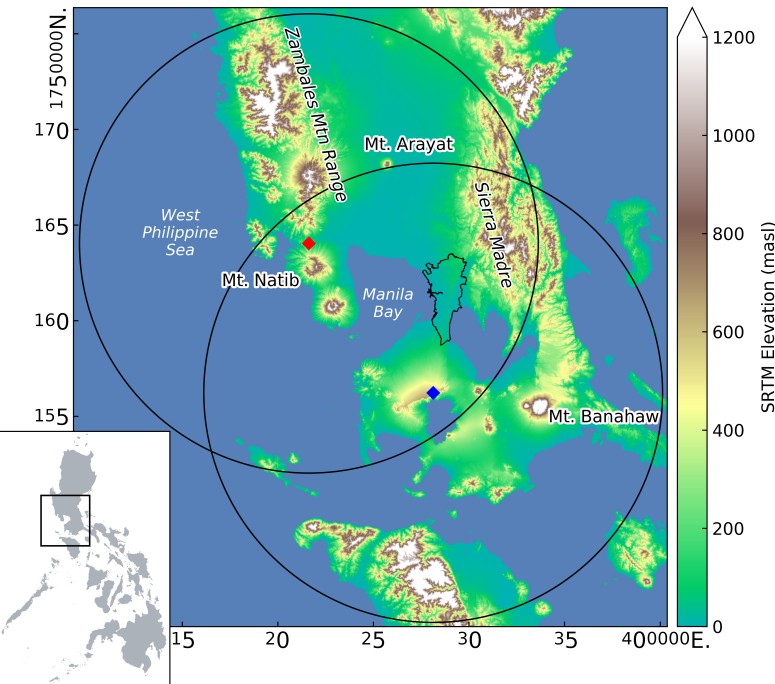

**Figure 1.** Locations of the SUB (red diamond) and TAG (blue diamond) radars showing the 120 km range with the region of overlap. Metropolitan Manila is outlined in black beside Manila Bay. The relative location of the study area with respect to the Philippines is shown in the inset.

with a population of approximately 13 Million. This region of overlap regularly experiences torrential rains from monsoon and typhoons extending for several days (Heistermann et al., 2013a; Lagmay et al., 2015).

## 2.1 Subic radar (SUB)

The SUB radar is a single-polarization S-band radar situated on top of a hill at 532 m above sea level (a.s.l.) in the municipality of Bataan (location: 14.82°N, 120.36°E) (see Figure 1). To its north lies the Zambales Mountains (highest peak: 2037 m a.s.l.) and to its south stands Mt. Natib (1253 m a.s.l.). The Sierra Madre Mountains run along the eastern part of the Luzon Island, at the far-east end of the radar coverage. Technical specifications are given in Table 1. Please note that SUB sweeps at 1.5 and 2.4 degree elevation were excluded for the years 2013 and 2014, due to apparently erratic and inconsistent behaviour.

## 2.2 Tagaytay radar (TAG)

Located about 100 km across the Manila Bay from the SUB radar is the TAG radar, a dual-polarized C-band radar. It sits on the Taal Volcano caldera ridge at 752 m a.s.l. in the municipality of Batangas. The radar coverage also includes the southern part of the Sierra Madre Mountains. Technical specifications are available in Table 1.

**Table 1.** Technical specifications of SUB and TAG Radar.

| | SUB Radar | TAG Radar |
|---|---|---|
| Frequency | S-Band | C-Band |
| Polarization | Single-pol | Dual-pol |
| Position (lat/lon) | 14.822°N 120.363°E | 14.123°N 120.974°E |
| Altitude | 532 m a.s.l. | 752 m a.s.l. |
| Maximum Range | | 120 km |
| Azimuth Resolution | | 1 ° |
| Gate length | | 500 m |
| Number of elevation angles | | 14 |
| Elevation angles | 0.5°, 1.5°, 2.4°, 3.4°, 4.3°, 5.3°, 6.2°, 7.5°, 8.7°, 10°, 12°, 14°, 16.7°, 19.5° | |
| Volume cycle interval | 8 minutes | 15 minutes |
| Transmit type | | Simultaneous |
| Start of operation | 2012 | 2012 |

Data during the rainy seasons of 2012-2014 and 2016 are used in this study. The scanning setup for TAG was experimentally changed during 2015 and reverted back in 2016. In order to ensure homogeneity in the GR intercomparison, we excluded the year 2015 from the analysis.

## 2.3 Space-borne precipitation radar

Space-borne radar data were collected from TRMM 2A23 and 2A25 version 7 (NASA, 2017) for overpass events in 2012-2014, and GPM 2AKu version 5A products (Iguchi et al., 2018) from 2014-2016, during the rainy season of June to December. The products include, among others, an attenuation correction of observed reflectivity (see e.g. Iguchi et al. (2009) for the TRMM precipitation radar; and Seto and Iguchi (2015) for GPM). Data were downloaded from NASA's Precipitation Processing System (PPS) through the STORM web interface (https://storm.pps.eosdis.nasa.gov/storm/). The parameters of TRMM/GPM extracted for the analysis (Table 2) are the same as those specified in Table 3 of Warren et al. (2018).

## 3 Methods

## 3.1 Overview

We facilitate the comparison of effectively three instruments: the two ground radars and the space-borne radar (see Section 2). While throughout the study period, the available space-borne radar platform changed from TRMM (2012-2014) to GPM (2014-2016), the consistency between the two for the year 2014 for the study area (Crisologo et al., 2018) allows us to consider the two rather as a single reference instrument. The comparison of the three platforms has two main components:

**Table 2.** TRMM and GPM parameters used for analysis, based on Table 3 of Warren et al. (2018)

| Satellite | Product | Parameter | Description |
|---|---|---|---|
| TRMM | 2A23 | dataQuality | Quality index for scan data |
| | | rainFlag | Flag indicating likelihood of surface precipitation in ray |
| | | rainType | Classification of precipitation in ray |
| | | HBB | Height of bright band (if present) in ray |
| | | BBwidth | Width of bright band (if present) in ray |
| | | status | Quality index for 2A23 products |
| | 2A25 | scLocalZenith | Zenith angle of ray at earth ellipsoid |
| | | correctZFactor | Attenuation-corrected reflectivity |
| GPM | 2AKu | dataQuality | Quality index for scan data |
| | | localZenithAngle | Zenith angle of ray at earth ellipsoid |
| | | flagPrecip | Flag indicating presence of precipitation in ray |
| | | heightBB | Height of bright band (if present) in ray |
| | | widthBB | Width of bright band (if present) in ray |
| | | qualityBB | Quality index for brightband identification |
| | | typePrecip | Classification of precipitation in ray |
| | | qualityTypePrecip | Quality index for precipitation type classification |
| | | zFactorCorrected | Attenuation-corrected reflectivity |

1. The SR-GR comparison is motivated by the estimation of the GR calibration bias. We define that bias as the mean difference ($\overline{\Delta Z}_{SR-SUB}$ or $\overline{\Delta Z}_{SR-TAG}$, in dBZ) between SR and GR, assuming SR to be a well-calibrated reference. As shown by Crisologo et al. (2018), we can improve the bias estimation if we give a lower weight to those matched samples which we assume to be affected by a systematic GR measurement error. Please note that we use the term *calibration bias* throughout the paper, as it is more commonly used. Strictly speaking, though, it is rather an *"instrument bias"* that lumps over any systematic effects of calibration and instrument stability along the radar receiver chain.

2. The GR-GR comparison is motivated by the evaluation of the consistency between the two ground radars. For that purpose, we can consider the mean difference ($\overline{\Delta Z}_{TAG-SUB}$) between the two ground radars (in dBZ) and the standard deviation of the differences ($\sigma(\Delta Z_{TAG-SUB})$, in dBZ). The differences in the region of overlap of two error-free ground radars would have a mean and a standard deviation of zero. Different levels of miscalibration of the two ground radars would increase the absolute value of the mean difference (which, in turn, implies that the mean difference would be zero if both GR were affected by the same level of miscalibration). But what about systematic measurement errors that are spatially heterogeneous in the region of overlap (such as beam blockage or PIA)? Although they could also affect the mean difference, we expect them to particularly increase the standard deviation of the differences. Hence, a removal of spatially heterogeneous measurements errors from both GR would reduce $\sigma(\Delta Z_{TAG-SUB})$, while a correc-

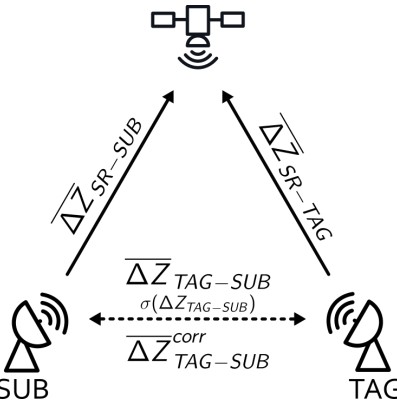

**Figure 2.** Schematic diagram of the SR-GR calibration bias estimation and GR-GR inter-comparison. The SUB and TAG calibration biases ($\overline{\Delta Z}_{SR-SUB}$ and $\overline{\Delta Z}_{SR-TAG}$, respectively) are calculated with respect to SR, and used to correct the ground radar reflectivities. The mean difference between SUB and TAG radars are calculated before ($\overline{\Delta Z}_{TAG-SUB}$) and after bias correction ($\overline{\Delta Z}_{TAG-SUB}^{corr}$)

tion of calibration bias of both GR would reduce the absolute value of $\overline{\Delta Z}_{TAG-SUB}$. And while we admit that neither $\overline{\Delta Z}_{TAG-SUB}$ nor $\sigma(\Delta Z_{TAG-SUB})$ could be considered imperative measures of reliability of any of the two ground radars, we still assume that any decrease in their absolute values would raise our confidence in any of the two radars' reflectivity observations.

## 3.2 SR–GR matching

To determine the calibration bias of each radar, we employ the relative calibration approach by using the space-borne-radar as a reference. In order to avoid introducing errors by interpolation, we use a volume-matching procedure. The 3D geometric matching method proposed by Schwaller and Morris (2011), further developed by Warren et al. (2018), was used to match SR bins to GR bins. This method has been implemented with the SUB radar for the same time period by Crisologo et al. (2018). In this study, we extend it to the TAG radar. Since the two radars are operating under the same scanning strategy and spatial resolution, the thresholds applied in filtering the data are kept the same as in the SR-SUB comparison described in Section 3.2, Table 2, of Crisologo et al. (2018), except that we considered samples only from below the bright band (as specified by the bright band detection in the SR product, see Table 2). That methodological adjustment was necessary due to the conversion between Ku and C-band reflectivity, which accounts for the systematic effect of different measurement frequencies: For that conversion, we used an empirical function published by Louf et al. (2019), Eq. 5, which was derived from T-matrix scattering simulations. According to the authors, that function is only valid for liquid rain; hence we excluded samples from within and above the bright band. The same was done for the S-band radar, in order to keep the matching procedure consistent between SUB and TAG. The conversion from Ku- to S-band reflectivity was implemented using the functions published by Cao et al. (2013). Further details of the SR data specifications and the matching procedure can be found in Crisologo et al. (2018).

### 3.3 GR–GR matching

We compare the reflectivities of both ground radars in the overlapping region to quantify the mean and the standard deviation of their differences, and thus the effectiveness of the quality-weighting and the relative calibration procedure. Please note that we do not explicitly account for differences in the reflectivity factor between S-band and C-band due to resonance effects, although Baldini et al. (2012) found that for very high reflectivities and very high median volume diameters of the drop size distribution, the deviation between the reflectivity factors of S-band and C-band can reach up to a maximum of 3 dB. Yet, we assume that, in such a scenario, the uncertainty introduced by path-integrated attenuation and its correction for C-band is more important, and at the same time implicitly addressed by the quality-weighting framework. In order to compare reflectivities from different radars, the different viewing geometries must be carefully considered. The polar coordinates of each radar are transformed into azimuthal equidistant projection coordinates, centered on each radar. Each radar cartesian coordinate is then transformed into the other radar's spherical coordinate system, such that each of the radar bins of the TAG radar have coordinates with respect to the SUB radar, and vice versa. For this purpose, we use the georeferencing module of the wradlib library (https://wradlib.org) which allows for transforming between any spherical and Cartesian reference systems. Bins of the TAG radar that are less than 120 km away from the SUB radar are chosen. The same is done for bins of the SUB radar. In order to match only bins of similar volume, Seo et al. (2014) suggested a matching zone of 3 km within the equidistant line between the two radars. We decided to make this requirement less strict in order to include more matches, and thus extended this range to 10 km. From the selected bins, each SUB bin is matched with the closest TAG bin, not exceeding 250 m in distance. The matching SUB and TAG bins are exemplarily shown in black in Figure 3 for the 0.5 degree elevation angle, such that each black bin in the SUB row corresponds with a black bin in the TAG row.

### 3.4 Estimation of path-integrated attenuation

Atmospheric attenuation depends on the radar's operating frequency (Holleman et al., 2006). For radar signals with wavelengths below 10 cm (such as C- and X-band radars), significant attenuation due to precipitation can occur (Vulpiani et al., 2006), depending on precipitation intensity (Holleman et al., 2006). In tropical areas such as the Philippines, where torrential rains and typhoons abound, C-band radars suffer from substantial PIA.

In this study, we did not correct the ground radar reflectivity for attenuation. Instead, we require PIA estimates as a quality variable to assign different weights of GR reflectivity samples when computing quality-weighted averages of reflectivity (see section 3.6). For that purpose, PIA is estimated by using dual-polarization moments observed by the TAG radar. The corresponding procedure includes the removal of non-meteorological echoes based on a fuzzy echo classification, and the reconstruction of the differential propagation phase from which PIA is finally estimated. The method is based on Vulpiani et al. (2012), and was comprehensively documented and verified for the TAG radar by Crisologo et al. (2014) which is why we only briefly outline it in the following.

The fuzzy classification of meteorological vs. non-meteorological echoes was based on the following decision variables: the Doppler velocity, the copolar cross-correlation, the textures (Gourley et al., 2007) of differential reflectivity, differential

propagation phase ($\Phi_{DP}$), and a static clutter map. The parameters of the trapezoidal membership functions as well as the weights of the decision variables are specified in Table 2 of Crisologo et al. (2014). Bins classified as non-meteorological were removed from the beam profile and then filled in the subsequent processing step. In that step, a clean $\Phi_{DP}$ profile is reconstructed by removing the effects of wrapping, system offset and residual artifacts. The reconstruction consists of an iterative procedure in which specific differential phase ($K_{DP}$) is repeatedly estimated from $\Phi_{DP}$ using a convolutional filter, and $\Phi_{DP}$ again retrieved from $K_{DP}$ via integration, after filtering spurious and physically implausible $K_{DP}$ values.

According to Bringi et al. (1990), specific attenuation, $\alpha_{hh}$ (dB km$^{-1}$), is linearly related to $K_{DP}$ by a coefficient $\gamma_{hh}$ (dB deg$^{-1}$) which we assume to be constant in time and space with a value of $\gamma_{hh} = 0.08$ (Carey et al., 2000). Hence, the two-way path-integrated attenuation, $A_{hh}$ (dB), can then be obtained from the integral of the specific attenuation along each beam - which is equivalent to our reconstructed $\Phi_{DP}$ from which the system offset ($\Phi_{DP}(r_0)$) was removed in the previous step.

$$A_{hh}(s) = 2 \int_{r_0}^{r} \alpha_{hh}(s) ds \tag{1}$$

$$= 2\gamma_{hh} \int_{r_0}^{r} K_{DP}(s) ds \tag{2}$$

$$= \gamma_{hh} (\Phi_{DP}(r) - \Phi_{DP}(r_0)) \tag{3}$$

### 3.5 Beam Blockage

In regions of complex topography, the ground radar beam can be totally or partially blocked by topographic obstacles, resulting in weakening or loss of the signal. To simulate the extent of beam blockage for each ground radar, as introduced by topography, we used the algorithm proposed by Bech et al. (2003), together with the Shuttle Radar Topography Mission (SRTM) Digital Elevation Model (DEM) with a 1 arc-second (approximately 30 m) resolution. The procedure has been documented in Crisologo et al. (2018) in more detail. In summary, the values of the DEM are resampled to the radar bin centroid coordinates to match the polar resolution of the radar data. Then, the algorithm computes the beam blockage fraction for each radar bin by comparing the elevation of the radar beam in that bin with the terrain elevation. Finally, the cumulative beam blockage fraction (BBF) is calculated for all the bins along each ray, where a value of 1.0 corresponds to a total occlusion and a value of 0.0 to complete visibility.

### 3.6 Quality index and quality-weighted averaging

The quality index is a quantity used to describe data quality, represented by numbers ranging from 0 (poor quality) to 1 (excellent quality), with the objective of characterizing data quality independent of the source, hardware, and signal processing (Einfalt et al., 2010).

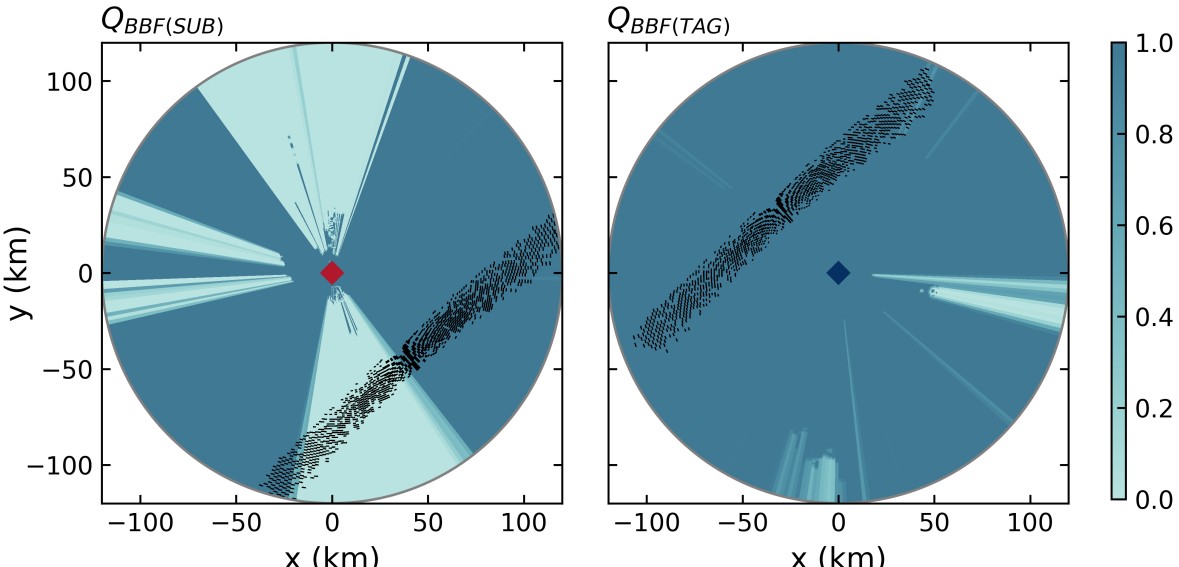

**Figure 3.** The beam blockage quality index ($Q_{BBF}$) for the two radars is shown in the background for $0.5°$ elevation angle. Black points show the locations of matched bins between SUB and TAG for each radar coverage, exemplarily for an elevation angle of $0.5°$.

To calculate a quality index for the beam blockage fraction, the transformation function suggested by Zhang et al. (2011) is used:

$$Q_{BBF} = \begin{cases} 1 & BBF \leq 0.1 \\ 1 - \frac{BBF - 0.1}{0.4} & 0.1 < BBF \leq 0.5 \\ 0 & BBF > 0.5 \end{cases} \tag{4}$$

Figure 3 shows the beam blockage quality index ($Q_{BBF}$) maps of SUB and TAG for the lowest elevation angle. SUB is
5    substantially affected by beam blockage in the northern and southern sector, due to the radar sitting between two mountains along a mountain range. The southern beam blockage sector of the SUB radar clearly affects the region of overlap with the TAG radar. Meanwhile, TAG has a clearer view towards the north, with only a narrow sector to the east and partially in the south being affected by a very high beam blockage. It is not shown in the figure, but the higher elevation angles ($> 0.5°$) of the TAG radar are not affected by any beam blockage.
10    For path-integrated attenuation, the values are transformed into a quality index as

$$Q_{PIA} = \begin{cases} 1 & \text{for} \quad A_i < A_{min} \\ 0 & \text{for} \quad A_i > A_{max} \\ \frac{A_{max} - A_i}{A_{max} - A_{min}} & \text{else,} \end{cases} \tag{5}$$

following the function proposed by Friedrich et al. (2006), where $A_{min}$ and $A_{max}$ are the lower and upper attenuation thresholds. The values for $A_{min}$ and $A_{max}$ are chosen to be 1 dB and 10 dB.

Multiple quality indices from different quality variables can be combined in order to obtain a single index of total quality. Different combination approaches have been suggested, e.g. by addition or multiplication (Norman et al., 2010), or by weighted averaging (Michelson et al., 2005). We chose to combine $Q_{BBF}$ and $Q_{PIA}$ multiplicatively, in order to make sure that a low value of either of the two propagates to the total quality index ($Q_{GR} = Q_{GR,BBF} * Q_{GR,PIA}$).

It should be noted that $Q_{SUB,PIA}$ is always considered to have a value of 1, as we consider attenuation negligible for S-band radars, so that effectively $Q_{SUB} = Q_{SUB,BBF}$.

Based on this quality index $Q_{GR}$, we follow the quality-weighting approach as outlined in Crisologo et al. (2018). For each match between SR and GR bins, the quality $Q_{match}$ is obtained from the minimum $Q_{GR}$ value of the GR bins in that match. We then compute the average and the standard deviation of the reflectivity differences between SR and GR by using the $Q_{match}$ values as linear weights (see Crisologo et al. (2018) for details). We basically follow the same approach when we compute the quality-weighted average and standard deviation of the differences between the two ground radars, SUB and TAG, in the region of overlap. Here, the quality $Q_{match}$ of each match is computed as the product $Q_{SUB} * Q_{TAG}$ of the two matched GR bins.

It should be emphasized at this point that, in the region of overlap, the TAG radar is not affected by beam blockage. So while the computation of calibration bias for the TAG radar, based on SR overpasses, is affected by $Q_{TAG,BBF}$ (as it uses the full TAG domain), the comparison of SUB and TAG reflectivities is, in fact, only governed by $Q_{SUB,BBF}$ and $Q_{TAG,PIA}$.

### 3.7 Computational details

Following the guidelines for transparency and reproducibility in weather and climate sciences as suggested by Irving (2016), we have made the entire processing workflow and sample data available online at https://github.com/IreneCrisologo/inter-radar (Crisologo and Heistermann, 2020). The main components of that workflow are based on the open source software library for processing weather radar data called wradlib (Heistermann et al., 2013b), version 1.2 (released on 31.10.2018) based on Python 3.6. The main dependencies of wradlib include Numerical Python (NumPy; Oliphant (2015), Matplotlib (Hunter, 2007), Scientific Python (SciPy; Virtanen et al. (2019)), h5py (Collette, 2013), netCDF4 (Rew et al., 1989), gdal (GDAL Development Team, 2017), and pandas (McKinney, 2010).

### 4 Results and Discussion

The presentation and discussion of results falls into four parts.

1. In section 4.1, we demonstrate the effect of extending the framework of quality-weighting by path-integrated attenuation. This is done by analysing the mean and the standard deviation of differences between the two ground radars, SUB and TAG, in different scenarios of quality filtering for a case in December 2014.

2. In section 4.2, we construct a time series of calibration bias estimates for the TAG C-band radar by using the extended quality-averaging framework together with space-borne reflectivity observations from TRMM and GPM overpass events. This time series complements the calibration bias estimates we had already gathered for the SUB S-band radar in Crisologo et al. (2018).

3. In section 4.3, we use the calibration bias estimates for SUB and TAG in order to correct the GR reflectivity measurements, and investigate whether that correction is in fact able to reduce the absolute value of the mean difference $\overline{\Delta Z}_{TAG-SUB}$ between the two radars. This analysis is done for events in which we have both valid SR overpasses for both radars and a sufficient number of samples between the two ground radars in the region of overlap.

4. In section 4.4, finally, we evaluate different techniques to interpolate the sparse calibration bias estimates in time, attempting to correct ground radar reflectivity observations also for times in which no overpass data is available. The effect of different interpolation techniques is again quantified by the mean difference $\overline{\Delta Z}_{TAG-SUB}$ between the two ground radars.

## 4.1 The effect of extended quality filtering: the case of December 9, 2014

In this section, we demonstrate the effect of extending the quality framework by path-integrated attenuation. In Figure 3, we have already seen that the SUB radar is strongly affected by beam blockage in the region of overlap. Yet, as an S-band radar, it is not significantly affected by attenuation. For the TAG radar, it is vice versa: not much affected by beam blockage, yet it will be affected by atmospheric attenuation during intense rainfall. That setting provides an ideal environment to experiment with different scenarios of quality filtering. For such an experiment, we chose a heavy rainfall event on December 9, 2014, where there are more than 900 radar bins with precipitation in the region of overlap. The scan times are 06:55:14 and 06:57:58 (local times) for the SUB and TAG radars, respectively.

Figure 4 shows scatter plots of matched reflectivities in the region of overlap, combining matched GR bins from all elevation angles. Note that in this region of overlap, $Q_{SUB}$ is equivalent to $Q_{BBF}$, and $Q_{TAG}$ is dominated by $Q_{PIA}$. To illustrate the individual effects of the quality indices in the comparison, we simply refer to the dominating quality index instead of the associated radar (i.e. $Q_{BBF}$ for SUB and $Q_{PIA}$ for TAG). The points in the scatter plot are colored depending on the quality index of the corresponding matched sample: in Figure 4a, we can see that matches with a very low $Q_{BBF}$ value (i.e. high beam blockage) are concentrated above the 1:1 line, since beam blockage causes the SUB radar to underestimate in comparison to the TAG radar. If we consider each matched sample irrespective of data quality, the mean difference between the two radars is 1.7 dB, with a standard deviation of 8.1 dB. Taking $Q_{BBF}$ into account changes the mean difference to -1.9 dB—which is higher in absolute terms—and decreases the standard deviation to 5.5 dB. Figure 4b demonstrates the effect of using only PIA for quality filtering: Points with low $Q_{PIA}$ (i.e. high PIA) are concentrated below the 1:1 line, corresponding to an underestimation by the TAG radar as compared to the SUB radar. Considering only $Q_{PIA}$ for quality-weighting increases the mean difference between TAG and SUB to a value of 3.5 dB, and decreases the standard deviation just slightly to a value of 7.5 dB. By combining the two quality factors, we can reduce the absolute value of $\overline{\Delta Z}_{TAG-SUB}$ from 1.7 dBZ to -0.7 dB,

and, more notably, the standard deviation from 8.1 dBZ to 4.6 dBZ (Figure 4c). That effect also becomes apparent in Figure 4d in which we show how the multiplicative combination of quality factors not only pushes the mean of the differences towards zero, but also narrows down the distribution of differences dramatically.

Considering component (2) from section 3.1, it is the reduction of standard deviation that we are most interested in at this point: it demonstrates that the two GR become more consistent if we filter systematic errors that are spatially heterogeneous in the region of overlap. The low absolute value of the mean difference is, for this case study, not a result of correcting for calibration bias—which is addressed in the following sections.

On the basis of these results, we will, in the following sections, only refer to values of mean and standard deviation of SR-GR or GR-GR differences that are computed by means of quality-weighting, with the quality of a matched sample quantified as $Q_{match}$.

## 4.2   Estimating the GR calibration bias from SR overpass events

In Figure 8a of Crisologo et al. (2018), we had already shown the time series of quality-averaged differences between the SUB ground radar and the SR platforms TRMM and GPM, using beam blockage as a quality variable. For this study, we recomputed these values after excluding samples from above the bright band (please see section 3.2 for further explanation). Extending the framework for quality-weighted averaging by PIA, we have now computed the corresponding time series of quality-weighted mean differences for the TAG radar. Figure 5 shows the time series of calibration biases, as estimated from quality-weighted mean differences, for both SUB and TAG radars for years 2012-2014 and 2016. The first panel corresponds to Figure 8a of Crisologo et al. (2018). For SUB, there is a total of 95 SR overpass events that fit the filtering criteria referred to in Section 3.2, while for TAG, we only found 45 matches. Compared to the space-borne radars, both SUB and TAG are dramatically underestimating at the beginning of operation in 2012, where the underestimation of the TAG radar is even more pronounced. In 2014, the calibration improves for both radars significantly, and reaches an optimum (with regard to both radars) in 2016.

As pointed out in Crisologo et al. (2018), there is a strong variability of the estimated calibration biases between overpasses for SUB. This behaviour can be confirmed for the TAG radar, with particularly severe cases in 2013. Potential causes for this short-term variability have been discussed in Crisologo et al. (2018), and could include, e.g., residual errors in the volume sample intersections, short-term hardware instability, rapid changes in precipitation during the time interval between GR sweep and SR overpass, and uncertainties in the estimation of PIA, to name a few.

## 4.3   The effect of bias correction on the GR consistency: case studies

In this and the following section, we evaluate the effect of using the calibration bias estimates obtained from SR overpasses to actually correct the GR reflectivity measurements. We start, in this section, by analysing events in which we have both: valid SR overpass events for SUB and TAG, as well as at least 30 matched GR samples in the region of overlap. In that way, we can directly evaluate how an "instantaneous" estimate of the GR calibration bias estimates affects the GR consistency, as explained in component (2) of section 3.1. In contrast to section 4.1, where we focused on the standard deviation of differences between the two ground radars, we now focus on the mean differences in order to capture the effect of bias correction.

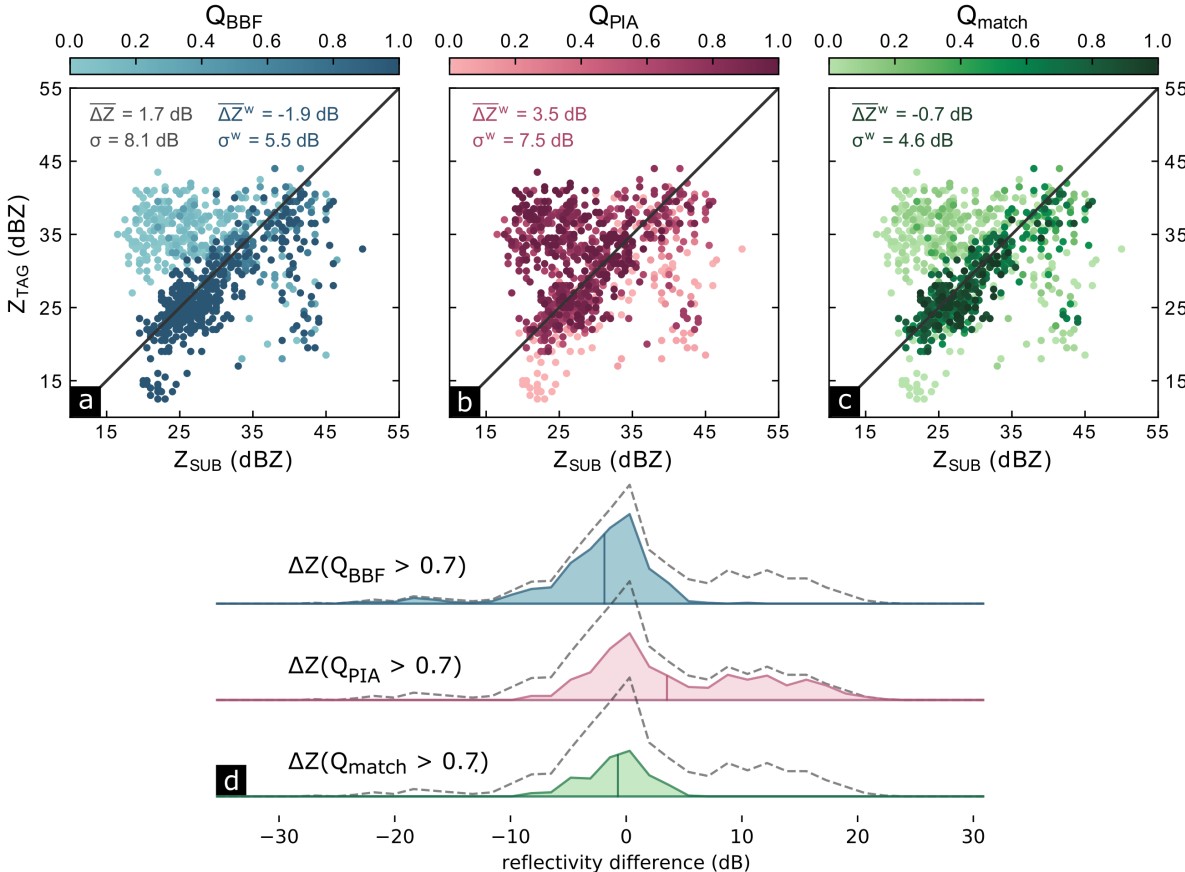

**Figure 4.** Scatter plot of reflectivity matches between TAG and SUB radars. The marker color scale represents the data quality based on (a) beam blockage fraction ($Q_{BBF}$), (b) path-integrated attenuation ($Q_{PIA}$), and (c) the multiplicative combination of the two ($Q_{match}$), where the darker colors denote high data quality and lighter colors signify low data quality. The ridgeline plots (d) show the distribution of the reflectivity differences of the remaining points if we choose points only with high quality indices (in this case, we select an arbitrary cutoff value of $Q_{match} = 0.7$). The mean is marked with the corresponding vertical line. The dashed lines represent the distribution of reflectivity differences of all points, when no filter is applied.

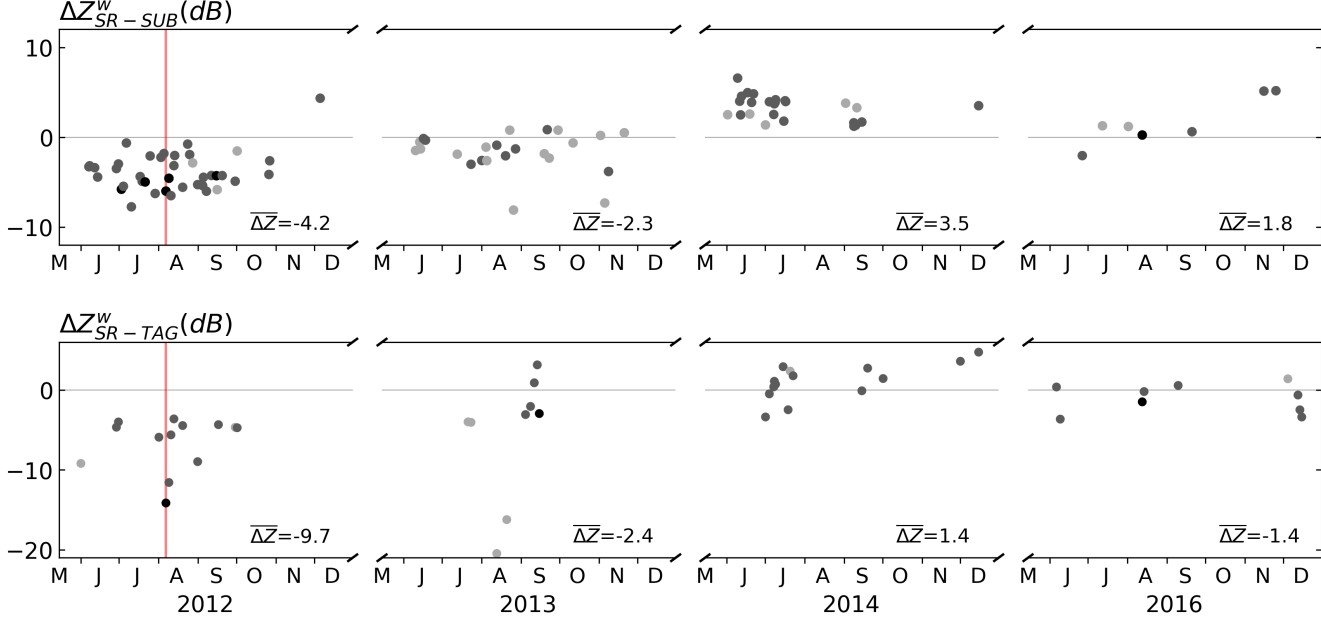

**Figure 5.** Calibration biases derived from comparison of GR with SR for SUB (a) and TAG (b) for the wet seasons (June to December) of the entire dataset. Symbols are coloured according to the number of matched samples: light grey: 10–99, medium grey: 100–999, and black: 1000+. The red line marks 06 August 2012 for the case study presented in Figure 6.

The first case is a particularly illustrative example: an extreme precipitation event that occurred in the region of overlap at a time in which both radars, SUB and TAG, apparently were affected by large miscalibration (see Figure 6) during the so-called *Habagat of 2012*, an enhanced monsoon event that happened in August 2012 (Heistermann et al., 2013a).

Figure 6a and b illustrate the estimation of the calibration bias for the SUB and TAG radars from TRMM overpass data. The calibration bias estimates of -5.1 dB (for SUB) and -14.1 dB (for TAG) obtained from those scatter plots correspond to the dots intersecting the red line in the time series shown in Figure 5. Figure 6c shows the matching reflectivity samples of the two ground radars, SUB and TAG, in the region of overlap which have *not yet* been corrected for calibration bias. The quality-weighted mean difference of reflectivies amounts to -12.2 dB. Accordingly, Figure 6d shows the matches in the region of overlap, with both SUB and TAG reflectivities corrected for calibration bias, based on the values obtained from Figure 6a and b, respectively. The corresponding value of the mean difference amounts to -3.4 dB. These effects are further illustrated by Figure 6e which shows the distributions of SR-GR and GR-GR differences before and after bias correction.

The case clearly demonstrates how large levels of miscalibration (-5.1 and -14.1 dB) can be reduced if an adequate SR overpass is available. That is proved by the large reduction of the absolute value of mean difference between the two ground radars, or, inversely, the large gain in GR consistency. Yet, the bias could not be entirely eliminated, which suggests that other systematic sources of error have not been successfully addressed for this case.

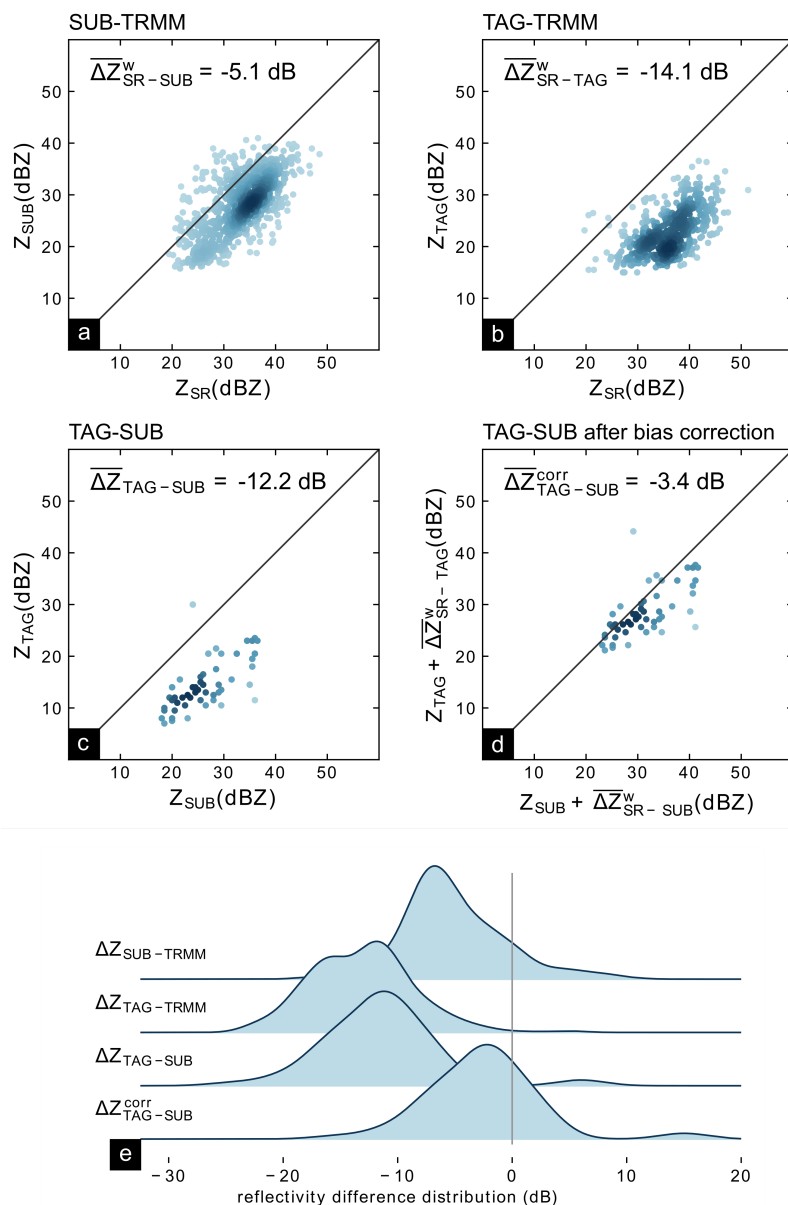

**Figure 6.** 3-way case study for 2012-08-06 17:15:47 local time. (a) and (b): Scatter plots of SR-GR comparisons between TRMM and SUB and TAG radars for points where $Q_{match}$>0.7, where the darkness of the color represents the point density. The corresponding weighted biases are calculated for each radar. (c) and (d): GR-GR inter-radar consistencies before and after bias correction. (e) Distribution of the differences of the reflectivity pairs for each comparison scenario.

**Table 3.** Calibration biases and inter-radar consistencies for different bias calculation scenarios

| | number of points | $\overline{\Delta Z}^{w}_{SR-SUB}$ | $\overline{\Delta Z}^{w}_{SR-TAG}$ | $\overline{\Delta Z}^{nocorr}_{TAG-SUB}$ | $\overline{\Delta Z}^{w,corr}_{TAG-SUB}$ |
|---|---|---|---|---|---|
| 2012-06-11 21:37:41 | 528 | -3.0 | -6.3 | -3.5 | 0.1 |
| 2012-06-28 22:14:46 | 48 | -3.3 | -4.7 | -1.3 | -0.4 |
| 2012-07-02 20:09:47 | 1248 | -5.9 | -11.5 | -7.0 | -3.0 |
| 2012-08-06 17:17:23 | 1121 | -5.1 | -14.1 | -12.2 | -3.4 |
| 2012-08-31 13:44:31 | 34 | -5.3 | -9.0 | -1.9 | 0.7 |
| 2016-08-12 11:40:27 | 1277 | 1.0 | -1.5 | -5.7 | -3.8 |

Table 3 summarizes our analysis of five additional events in which valid SR overpasses for both SUB and TAG coincided with a significant rainfall in the region of overlap between the two ground radars, most of which took place in 2012 (and one in 2016). Columns $\overline{\Delta Z}^{w}_{SR-SUB}$ and $\overline{\Delta Z}^{w}_{SR-TAG}$ show varying levels of calibration bias for SUB and TAG, quantified by the quality-weighted mean difference to the SR observations, together with varying levels of mismatch between the two ground radars, as shown by column $\overline{\Delta Z}^{nocorr}_{TAG-SUB}$. Using the calibration bias estimates for correcting the GR observations, we consistently reduce the quality-weighted mean difference between both ground radars, as expressed by column $\overline{\Delta Z}^{w,corr}_{TAG-SUB}$.

Altogether, the correction of GR reflectivity with calibration bias estimates of SR overpasses improves the consistency between the two ground radars which have shown largely incoherent observations *before* the correction. In all cases (including the *Habagat of 2012*), we were able to reduce the mean difference between the ground radars.

The question now is: Can we use these sparse calibration bias estimates also for points in time in which no adequate SR overpass data are available? Or, in other words, can we interpolate calibration bias estimates in time?

### 4.4 Can we interpolate calibration bias estimates in time?

The SR platform rarely overpasses both GR radar domains in a way that significant rainfall sufficiently extends over both GR domains including the GR region of overlap. Hence, our previous demonstration of the effective correction of GR calibration bias yielded only a few examples. From a more practical point of view, however, we are more interested in how we can use SR overpass data for those situations in which adequate SR coverage is unavailable—which is, obviously, rather the rule than the exception. An intuitive approach is to interpolate the calibration bias estimates from valid SR overpasses in time, and use the interpolated values to correct GR observations for any point in time. We can do such an interpolation independently for each ground radar, based on the set of valid SR overpasses available for each. In order to examine the effectiveness of such an interpolation, we again use the absolute value of the mean difference between the two ground radars as a measure of their (in-)consistency. Based on the reduction of that absolute value, as compared to uncorrected GR reflectivities, we benchmark the performance of three interpolation approaches:

1. *Linear interpolation* in time;

2. *Moving average*: we compute the calibration bias at any point in time based on calibration bias estimates in a 30-day window around that point, together with a triangular weighting function;

3. *Seasonal average*: For any point in time in the analyzed wet season of a year, we compute the calibration bias as the average of all calibration bias estimates available in that year.

This benchmark analysis is not considered to be comprehensive, but rather exemplary in terms of examined interpolation techniques. The three techniques illustrate different assumptions on the temporal representativeness of calibration bias estimates, as obtained from SR overpasses: a *seasonal average* reflects a rather low level of confidence in the temporal representativeness. The underlying assumption would be that we consider any short-term variability as "noise" which should be averaged out. The linear interpolation puts more confidence into each individual bias estimate, and assumes that we can actually
interpolate between any two points in time. Obviously, a 30-day moving average is somewhere in between the two.

**Table 4.** Mean absolute $\Delta Z_{TAG-SUB}$ for different correction scenarios and years

| | Mean absolute $\Delta Z_{TAG-SUB}$ (dB) | | | |
|---|---|---|---|---|
| | No correction | Seasonal mean | Linear interpolation | Moving average |
| All years | 4.7 | 3.0 | 2.6 | 2.4 |
| 2012 | 4.7 | 2.6 | 2.4 | 2.1 |
| 2013 | 7.9 | 7.7 | 5.0 | 4.9 |
| 2014 | 2.9 | 2.1 | 1.8 | 1.7 |
| 2016 | 4.3 | 1.6 | 2.1 | 2.0 |

    Table 4 provides an annual summary of the mean absolute differences in reflectivity between the two ground radars, without bias correction and with correction of bias obtained from different interpolation techniques. Most importantly, the mean absolute difference between the radars is always lower *after* correction, irrespective of the year or the interpolation method. Hence, it appears generally preferable to use interpolated calibration bias estimates to correct GR reflectivities, instead of not
correcting for bias—even for those periods in which no valid SR overpasses are available. In total, the 30-day moving average slightly outperforms the other two interpolation methods; only in 2016, the seasonal average performs best. The performance of the moving average suggests that it is possible for the calibration of radars to drift slowly in time, with variability stemming from sources which are yet difficult to disentangle. It is also worth mentioning that, for 2016, the mismatch between SUB and TAG before bias correction is quite high (4.3 dB). That is not expected since the calibration of both radars appears to have
improved over time (see section 4.2 and Figure 5). So while the bias correction clearly improves the GR consistency in 2016 (e.g. to a value of 1.6 dB when using a seasonal average for interpolation), we have to suspect that other sources of uncertainty, together with the effect of limited samples sizes, affect the comparison of the two ground radars: e.g. uncertainties in beam propagation, or residual errors in the quantification of path-integrated attenuation and beam blockage.

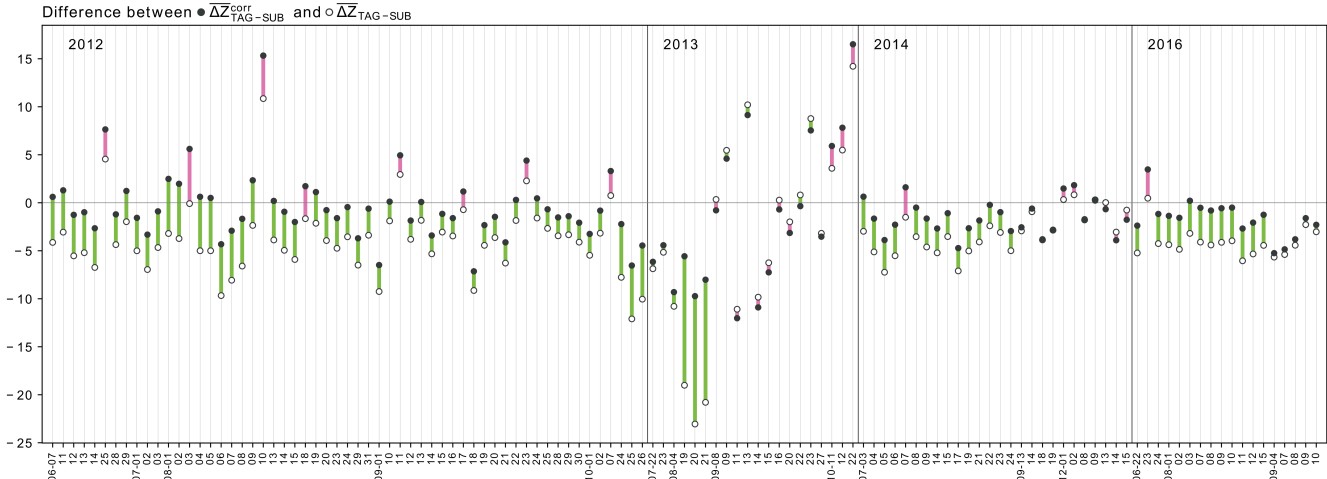

**Figure 7.** The differences between the inter-radar consistency before and after correcting for the ground radar calibration biases following a rolling window averaging for GR-GR pairs with more than 100 matches. The hollow (filled) circles represent the daily mean before (after) correction. The line color represents an improvement (green) or a decline (pink) in the consistency between the two ground radars.

In order to better understand the variability "behind" the annual averages in Table 4, Figure 7 shows the effects of bias correction on a daily basis, exemplified for the moving average interpolation. The hollow circles represent the daily mean differences between the two ground radars before ($\overline{\Delta Z}_{TAG-SUB}$) correction, while the filled circles show the daily mean differences after ($\overline{\Delta Z}_{TAG-SUB}^{w,corr}$) correction. The length of the bar shows the magnitude of the change, while the color of the

5   bar signifies improvement or degradation of consistency between the ground radars. A green bar denotes that the absolute value of the mean difference after correction has decreased, ı.e. the mean difference after correction (filled circles) is closer to zero than before correction (unfilled circles). A pink bar denotes an increase in the absolute value of mean difference between the two radars after correction. In 82 out of 121 days, bias correction improves the consistency between the two ground radars by more than 1 dB. Inversely, though, this implies that in 18 out of 121 days, the use of interpolated bias estimates causes

10   a degradation of consistency between the ground radars, expressed as an increase of more than 1 dB in the absolute mean differences. However, we are also able to identify several days for which the bias correction decreased the absolute mean differences, yet still not to a level that could be considered as acceptable for quantitative precipitation estimation.

## 5   Conclusions

Schwaller and Morris (2011) had presented a technique to match reflectivity observations from space-borne radars (SR) and

15   ground radars (GR). Crisologo et al. (2018) extended that technique by introducing the concept of quality-weighted averaging of reflectivity in order to retrieve the GR calibration bias from matching SR overpass data. They exemplified the concept of

quality weighting by using beam blockage as a quality variable, and demonstrated the effectiveness of the approach for the Subic (SUB) S-band radar in the Philippines.

The present study has extended the concept of quality-weighted averaging by accounting for path-integrated attenuation (PIA) as a quality variable, in addition to beam blockage. Accounting for PIA becomes vital for ground radars that operate at C- or X-band. In addition to the SUB S-band radar, this study has included the TAG C-band radar which substantially overlaps with the SUB radar.

In the first part of this study, we have demonstrated that only accounting for both, beam blockage and path-integrated attenuation, allows for a consistent comparison of observations from the two ground radars, SUB and TAG: after transforming the quality variables "beam blockage fraction" and "path-integrated attenuation" into quality indices $Q_{BBF}$ and $Q_{PIA}$, with values between zero and one, we computed the quality-weighted standard deviation of matching reflectivities in the region of overlap between the two ground radars for an event on December 9, 2014. Using a quality index based on the multiplicative combination of $Q_{BBF}$ and $Q_{PIA}$, we were able to dramatically reduce the quality-weighted standard deviation from 8.1 dBZ to 4.6 dBZ, while using $Q_{BBF}$ and $Q_{PIA}$ alone would have only reduced the standard deviation to 5.5 or 7.5 dBZ, respectively. Based on that result, we have used, with confidence, the combined quality index throughout the rest of the study.

The next step involved the retrieval of the GR calibration bias from SR overpass data for the TAG C-band radar (for the SUB S-band radar, that had already been done by Crisologo et al. (2018)). For each matched volume in the SR-GR intersection, the combined quality index was computed for the TAG radar, and used as weights in calculating the calibration bias as a quality-weighted average of the differences between SR and GR reflectivities. We applied this approach throughout a 4-year period to come up with a time series of the historical calibration bias estimates of the TAG radar, and found the calibration of the TAG radar to be exceptionally poor and volatile in the years 2012 and 2013, with substantial improvements in 2014 and 2016.

In order to demonstrate the effectiveness of estimating and applying the GR calibration bias obtained from SR overpass data, we have compared, in the region of overlap, the corrected and uncorrected reflectivities of the SUB and TAG radars, for six significant rainfall events in which all three instruments—TAG, SUB and the SR—had recorded a sufficient number of observations. We have shown that the independent bias correction is able to largely increase the consistency of the two ground radar observations, as expressed by a reduction of the absolute mean difference between the GR observations in the region of overlap, for each of the six events—in one case even by almost 8.9 dB. The main finding from these cases is, that we can legitimately interpret the quality-weighted mean difference between SR and GR reflectivities as the instantaneous GR calibration bias, even if the magnitude of that bias varies substantially within short periods of time.

Yet, the question remains how to correct for calibration bias in the absence of useful SR overpasses. That question is particularly relevant for the reanalysis of archived measurements from single-polarization weather radars. In this study, we have evaluated three different approaches to interpolate calibration bias estimates from SR overpass data in time: i) linear interpolation, ii) a 30-day moving average, and iii) a seasonal average. Each of these approaches illustrates different assumptions on the temporal representativeness of the calibration bias estimates. On average, all of these approaches produced calibration bias estimates that were able to reduce the mean absolute difference between the GR observations, which increases our confidence in the corrected GR observations. Of all interpolation approaches, the moving 30-day window outperformed the other two

approaches. However, we found that behind the average improvement of GR-GR consistency, there were also a number of cases in which the consistency between the ground radars was degraded, or in which high inconsistencies could not be significantly improved. Altogether, it still appears difficult to interpolate such a volatile behaviour, even if we considered the actual calibration bias estimates from the SR overpasses as quite reliable. A way to further investigate that behaviour would be to complement the analysis by relative calibration techniques that use ground clutter returns as a reference (e.g. Silberstein et al. (2008)). Although such techniques only allows to detect changes in calibration relative to a baseline, they can be applied to each volume cycle and thus inform us about dynamics at a high temporal resolution and coverage. That way, we could support the interpolation of bias estimates obtained from SR overpasses, or scrutinize the temporal variability of such estimates. An application and in-depth discussion of this concept has just recently been provided by Louf et al. (2019) with the example of the C-band weather radar in Darwin, Australia.

In that context, maintenance protocols of the affected ground radars would be very helpful in interpreting and interpolating time series of calibration bias estimates. Such records were unavailable for the present study, which made it difficult to understand the observed variability of calibration bias estimates. Yet, this information will mostly be internally available at those institutions operating the weather radars. With the software code and sample data of our study being openly available (Crisologo and Heistermann, 2020), such institutions are now enabled to carry out the analysis presented in this study by themselves, while being able to benefit from cross-referencing the results with internal maintenance protocols.

The correction of GR calibration appeared particularly effective during periods with large levels of miscalibration. For such cases, interpolated bias estimates allowed for an effective improvement of raw GR reflectivities. Yet, we need to continue disentangling different sources of uncertainty for both SR and GR observations in order to distinguish actual variations in instrument calibration and stability from measurement errors that accumulate along the propagation path, and to better understand the requirements to robustly estimating these properties from limited samples. That also includes to extend the quality-weighting framework to the quality of SR reflectivity measurements, as already outlined in Crisologo et al. (2018), in particular with regard to the combined effects of attenuation at Ku band and nonuniform beam filling which several authors found to cause systematic errors of SR reflectivity measurements in convective situations (see e.g. Munchak (2018); Deo et al. (2018) and Park et al. (2015) for an in-depth discussion). Progress on these ends should also improve the potential for interpolating calibration bias estimates in time, in order to tap the potential of historical radar archives for radar climatology, and to increase the homogeneity of composite products from heterogeneous weather radar networks.

*Code and data availability.* Code and sample data can be accessed via https://github.com/IreneCrisologo/inter-radar (Crisologo and Heistermann, 2020)

*Author contributions.* IC and MH conceptualized, prepared the code, performed the analysis, and wrote the manuscript.

*Competing interests.* The authors declare that they have no conflict of interest.

*Acknowledgements.* The radar data for this analysis were provided by the Philippine Atmospheric, Geophysical and Astronomical Services Administration (PAGASA, http://pagasa.dost.gov.ph). The study was also funded by the German government through the German Academic Exchange Service (https://www.daad.de/en/).

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
