# Peer review of "Using ground radar overlaps to verify the retrieval of calibration bias estimates from space-borne platforms"

_Atmospheric Measurement Techniques, 2019_

## Referee Comment (RC1) · Anonymous Referee #1 · 24 May 2019

Review

Journal: Atmos. Meas. Tech. Discussion

Title: Using ground radar overlaps to verify the retrieval of calibration bias estimates from spaceborne platforms

Summary:

This manuscript presents a study that extends previous ground radar (GR) calibration using satellite-based radar (SR). S-band and C-band GRs near Manilla (Philippines) are compared with Ku-band SR from TRMM/GPM to obtain the reflectivity calibration offset of the GRs over the course of four years while taking into account the quality of the GR measurements. The study takes into account the beam blockage and attenuation quality of the GRs and derives a quality index for the GR measurements. As a result, the GR is better calibrated by the SR than without considering the quality and provides improved consistency between the two GRs.

Contribution and Concerns:

There have been several studies in recent years to examine SR calibration of GR over time (e.g., Warren et al. 2018; Crisologo et al. 2018; Biswas and Chandrasekar 2018). This novelty of this study is that it incorporates path integrated attenuation of the GR measurements to improve the effectiveness of the quality-weighted GR calibration method introduced by Crisologo et al. 2018. Also, this study provides a technique for determining the SR-based calibration of GR between the occurrences of satellite overpasses. The topic of this study is of interest to the radar community and others relying on GR measurements.

However, the study seems to have overlooked the fact that SR measurements are not perfect either and one could argue also require a similar quality weighting. Yes, the SR reflectivity measurements used in this paper have been corrected for attenuation and this correction is only an estimate. The 2A25 and 2ADPR algorithms rely on the surface-reference technique (Meneghini et al. 2000) and Hitschfeld-Borden method (Hitschfeld and Bordan 1954) to correct the SR measured reflectivity. These techniques can fail or provide poor estimates when multiple scattering and/or non-uniform beam filling may be present, which typically occurs within deep convective precipitation, even at Ku-band (Munchak 2018). As a result, the GR calibration offsets determined by this study may be in error, at least during intense convection. Therefore the authors must address this concern about the quality of the SR measurements, primarily when intense precipitation is included within the matched sample volumes. Therefore the authors must address this concern about the quality of the SR measurements, primarily when intense precipitation is included within the matched sample volumes.

Another thing that could use additional explanation is the results presented in Table 3. The GR becomes less biased with time, and the relative improvement amongst the three interpolation techniques decreases with time. It even seems that some optimal calibration is attained by 2016. The authors should expand on this and suggest plausible causes for these trends.

Recommendation: Minor revision

Minor Comments:

1) Table 1...indicated whether the transmit type of each radar (e.g., SHV or alternating H/V)
2) Section 2.3: NASA, 2017 reference is missing from bibliography
3) Section 2.3: Suggest expanding upon the parameters used from TRMM/GPM instead of simply referring to Table 3 of Warran et al. (2018)
4) Figure 3…clarify the vertical reference of the scans (e.g., what elevation angle or constant altitude)
5) Section 3.6…7$^{th}$ paragraph…subscript Qmatch?
6) Figure 4d…define the dashed line above the histograms.
7) Figure 7…"samples with significant number of matches"…how many is significant?

Biswas, S., and V. Chandrasekar, 2018: Cross-Validation of Observations between the GPM Dual-Frequency Precipitation Radar and Ground Based Dual-Polarization Radars. *Remote Sens.*, **10**, 1773, doi:10.3390/rs10111773.

Hitschfeld, W., and J. Bordan, 1954: Errors Inherent in the radar measurement of rainfall at attenuating wavelengths. *J. Meteorol.*, **11**, 58–67, doi:10.1175/1520-0469(1954)011<0058:EIITRM>2.0.CO;2.

Meneghini, R., T. Iguchi, T. Kozu, L. Liao, K. Okamoto, J. A. Jones, and J. Kwiatkowski, 2000: Use of the Surface Reference Technique for Path Attenuation Estimates from the TRMM Precipitation Radar. *J. Appl. Meteorol.*, **39**, 2053–2070, doi:10.1175/1520-0450(2001)040<2053:UOTSRT>2.0.CO;2.

Munchak, S. J., 2018: Remote Sensing of Precipitation from Airborne and Spaceborne Radar. *Remote Sensing of Aerosols, Clouds, and Precipitation*, Elsevier, 267–299.

---

## Referee Comment (RC2) · Anonymous Referee #2 · 27 Jul 2019

Review of "Using ground radar overlaps to verify the retrieval of calibration bias estimates from space-borne platforms" by Irene Crisologo and Maik Heistermann

Summary: This study uses ground radar overlaps, in Philippines, to verify the retrieval of calibration bias estimates from the TRMM PR. It extends the concept of quality-weighted averaging by including path integrated attenuation, in addition to beam blockage, as a quality index for an effective radar reflectivity comparison. While the concept of relative calibration is not new, the reviewer is much interested in the extended approach and verification using overlaps and think that this would be a useful contribution to radar calibration and TRMM/GPM literature. My suggestions on developing the manuscript are listed below.

Major Comments

The following major point needs to be addressed by the authors:
1. The line numbers are preferred to be continuous and clear in the entire manuscript otherwise referencing becomes difficult (see for example page 1, 2 and 3).
2. Figure 2 quality seems to be compromised which could be improved
3. In Section 4.3, line 5 "..the value of mean difference amounts to -4.6 dB" and the authors suggest this deviation possibly due to "..systematic sources.." (line 10). I was wondering if the inherent behaviour of analysing a subset of data from the wider data has been examined here. The overlap data is a subset of the wider SUB/TAG radar data and the bias estimates are obtained from the wider GR-SR data (and not just the overlap data). Maybe highlighting the overlap region data in the SUB-TRMM and TAG-TRMM before and after correction in Figure 6 could help in understanding this deviation, which means adding two more panels similar to (a) and (b) but for GR-SR bias corrected (highlighting the overlap region data). This would show the relative positon of the overlap data about the 1:1 line after the bias correction. While this would be an eyeball analysis, a better approach would be to do some statistical analysis for the overlap data after correction. The authors should check this for all the other cases.
4. The authors could also examine the nature of the precipitation types studied here (fraction of convective or stratiform) as this could significantly affect the bias correction estimates (underestimation or overestimation: for more detail see studies such as Park et al 2015 and Deo et al 2018). This could help particularly to explain lines 13 -15 (sec. 4.4) "……17 out of 121 days, …. an increase of more than 1dB in the absolute mean differences" (as given in Figure 7 also).
5. Use of abbreviations such as SUB, TAG, GR, SR, PIA and etc. needs to addressed- use it consistently throughout the manuscript otherwise it distracts a reader: As an example, in lines 13 – 18 (sec. 3.6) there is a combination of SUB, TAG, Subic, Tagaytay in just one paragraph.

A list of minor comments/suggestions is as follows:

1. Change "spaceborne" to "space-borne" throughout the manuscript

Section 1

2. Line 1 "…observations are the key…" remove "the". Same line add article "a" before "large" and "high".
3. Line 3, change "errors" to "error" in "The estimation errors.."
4. Two lines before line 5: change "..- let it be.." to "..be it.."
5. Line 5, introduce PIA abbreviation here and "On top..." is confusing, please rephrase.
6. Line 10, remove "…maybe surprising to some…"
7. Line 32, change "finally" to "recently" in "…and finally by Warren..."
8. Page 3, line 16, change "latter" to "last" in "The latter item…"
9. Rephrase line 24-27 (Page 3). Remove " : " add a "and" before "section 3" in "…data sets: section 3…". Line 26, add full stop after "… "bias estimation" and then "Section 4 will present…results followed by the conclusion in section 5".

Section 2

10. Line 29 (page 3), Rephrase to "The Philippine weather agency, known as Atmospheric….(PAGASA), maintains…"
11. Line 32-33 (page 3-4), replace "at" with "with" and "inhabitants" with "population" in "…area at approximately 13 Million inhabitants."
12. Figure 1: Make the x and y label fonts consistent.
13. Page 4 line 5, define "a.s.l" or use long format "above sea level" in "…532 m a.s.l"
14. Page 4, line 8, change "available" to "given".
15. Page 4, line 11, "Band" should be lower case in "..C-Band..."
16. Page 4, line 12, change to "…sits on"
17. For sec. 2.1. and 2.2, see comment 5 in "Major comments"
18. Page 5, line 11, rephrase "…collected..."
19. Page 5, line 12, remove "The" from "The data…"
20. Page 5, line 12, "..same as specified…" should be "…same as those specified.."

Section 3
21. Page 5, Line 3, "(see 2)" should be "(see section 2)"
22. Page 5, Line 7, remove bold emphasis of words or sentences which also applies to other sections.
23. Page 6, lines 22-23, check the usage of brackets.
24. Line 23, remove italicised emphasis of words or sentences which also applies to other sections. Could place them in quotation marks if explicitly required.
25. Section 3.4, page 7, line 7, "dual-pol"?
26. Page 8, line 13, delete repeated "copolar cross-correlation"
27. Page 8, line 14, delete "and" in "…and differential propagation phase …"
28. Page 8, line 18, define $K_{DP}$.
29. Page 8, line 30, define DEM.
30. Page 8, last sentence, add "a" before "total" in "… corresponds to total.." and also before "complete" at the end of the sentence
31. Figure 3, I believe "...corresponding elevation angle." should have been "…corresponding sweep angle.)
32. Page 9, line 17, add "a" before "…very high beam..".
33. Page 9, line 17, state which "…higher elevations.." Is it $> 0.5$ or $1.5$?
34. Page 10, Equation 5, define $K_{r,s}$

Section 4 and 5

35. Page 11, line 28, define how much is sufficient in "…sufficient radar bins".
36. Page 11, line 28, is the time LT or UTC?
37. Page 12, line 9, rephrase to "…underestimation by the TAG…"
38. Page 12, line 15.Rephrase "Remembering item (2)…" to " Considering component (2)…"
39. Page 12, Line 19, remove brackets in "… (SR-GR or GR-GR)…
40. Page 12, line 27, delete "is" from "The first panel is corresponds…"
41. Page 12, line 28, "Section 3.2" instead of "Section III.2"
42. Page 12, line 33, change "drastic" to "severe"
43. Figure 4, panel d, what do the dashed lines represent? Also add article "a" before "low" and "high"
44. Page 14, line 6 define "sufficient number". Also add "In" before "That" in "That way, we…"
45. Page 14, line 8, change "item (2)" to "component (2)". Also replace "…in which…" with "where".
46. Page 14, line 10, replace "…took place right…" with "occurred"
47. Page 14, line 11, rephrase "according to Figure 6" to "see Figure 6" and put in brackets. Also add "during" before "…the so called Habagat…".
48. Table 2, define Npts.
49. Page 15, line 7, replace "massive" with large. This also applies to other sections.
50. Page 15, line 20, rephrase to "The question now is…"
51. Page 15, line 25, add "a" before "…few examples".
52. Figure 6, is time LT or UTC? Also include a colour bar to show the scale density and add "respectively for" after "consistencies" in (c) and (d)
53. Page 18, lines 11-12, rephrase.
54.  Page 18, line 16, I believe you meant "unacceptable" rather than "acceptable".
55. Page 15, line 19, check referencing format i.e. for Schwaller and Morris.
56. Page 19, line 31, could use "…main finding" instead of "…main lesson"
57. Page 19, line 35, use full form in "single-pol"
58. Page 19, line 36, may be use numerals (i,ii and iii) to list the approaches
59. Page 19, last line, replace "any" with "all" in "On average, any.."
60. Page 20, line 8, delete one of the "also".
61. Page 20, line 10, replace "Altogether" with "Hence".
62. Page 20, line 13, replace "hard" with "difficult"
63. Page 20, lines 16-17 please rephrase the sentence after the website link as it is confusing.
64. Page 20, line 18, "..in periods" should be ("...during periods…"
65. Page 20, line 22, replace "…estimating…" with "estimate"

---

## Referee Comment (RC3) · Anonymous Referee #3 · 27 Jul 2019

Review

Journal: Atmos. Meas. Tech. Discussion

Title: Using ground radar overlaps to verify the retrieval of calibration bias estimates from spaceborne platforms

Authors: Irene Crisologo and Maik Heistermann

General comments

This study verifies estimates for calibration bias of ground (S- and C-bands) radars using spaceborne radars such as TRMM PR and GPM DPR (KuPR). To extend a methodology by Crisologo et al. (2018) for using radars at higher frequencies for C- and X-bands, the current study introduces a quality-control index as a function of PIA. The calibration biases of each ground radar are individually estimated from the spaceborne radars, which is evaluated by comparing between the calibrated ground radars. This study also examines temporal adjustment of the calibration change from only infrequent matchups between spaceborne and ground radars.

This study demonstrates an application for cross calibration of ground radars using spaceborne radars as a reference, which will contribute to a cross calibration of ground radars all over the world. I think the paper can be published after some corrections and edits.

Specific comments

1) In section 2, descriptions of used data are overall lacked. How to originally calibrate the ground radars? Don't some references and/or descriptions of the ground radars exist? How to treat/correct the precipitation attenuation for ground/spaceborne radars? Does the current study compare what parameters of radar reflectivity with or without the attenuation? This study mentions Crisologo et al. (2018) and Warren et al. (2018), but it makes readers to feel unkindly. Especially, data of the C-band ground radar because this study newly utilizes the data. Please describe appropriately. I suggest some references of attenuation-correction methods for spaceborne radar data as follows: Iguchi et al. (2009) and Seto and Iguchi (2015).

2) This study uses the two ground radars at frequencies of S- and C-bands and the spaceborne radars at a frequency of Ku-band. How to consider the difference in a frequency? From Crisologo et al. (2018), a conversion due to the Mie-scattering effect from Ku-band to S-band is empirically considered in this study. However, the conversion among S-, C-, and Ku-bands is not described anything. Is the residual difference of the two ground radars mixed with the frequency difference? Please describe explicitly.

3) This study tries a temporal adjustment of calibration changes for ground radars from only

infrequent matchups with spaceborne radars. The relative calibration adjustment with ground clutter (e.g., Silberstein et al. 2008, Louf e t al. 2019) is one of typical methods. Each of the relative calibration methods with the ground clutter and the matchup with spaceborne radar has its merits and demerits. Please discuss in the manuscript.

Technical corrections

1) Page 2 line 23: The sentence in a parenthesis after "*estimation errors (in…* " is too long, so I might be better to replace it. For examples: "The estimation errors are defined as retrieval errors of the precipitation rate… the radar reflectivity factor Z; then the errors are caused by…".

2) Page 2 line 4: I do not understand "let it be …". What does it mean?

3) Please specifically indicate the frequency or wavelength of the radars in Sections 2.1 and 2.2.

4) "Bandwidth" in Table 1 is wrong. I think "Frequency" is appropriate.

5) Page 7 line 27 and Page 15 first paragraph in Section 4.4: SR has been already defined at Page 2. Why did you redefine SR? In Section 5 (summary), I understood the redefinition as a refresh.

6) Page 7 line 33: I can not find Table 3 in Crisologo et al. (2018). Is it Table 2 in Crisolog et al. (2018)? Please indicate the correct number.

7) Table 2: -5 and -7 is should be -5.0 and -7.0 if the significant digit of those values is correct in this study.

8) Some references lack information such as URL (e.g. Iguchi et al. 2010, Jone et al. 2014). Please check https://www.atmospheric-measurement-techniques.net/for_authors/manuscript_preparation.html. Incidentally, Iguchi et al. (2010) is too old for a reference. Please update appropriately as follows: https://pmm.nasa.gov/resources/documents/gpmdpr-level-2-algorithm-theoretical-basis-document-atbd.

References

1) Crisologo, I., Warren, R. A., Mühlbauer, K., and Heistermann, M.: Enhancing the consistency of spaceborne and ground-based radar comparisons by using beam blockage fraction as a quality filter, Atmospheric Measurement Techniques, 11, 5223–5236, https://doi.org/https://doi.org/10.5194/amt-11-5223-2018, 2018.

2) Iguchi, T., T. Kozu, J. Kwiatkowski, R. Meneghini, J. Awaka, and K. Okamoto, 2009: Uncertainties in the Rain Profiling Algorithm for the TRMM Precipitation Radar. J. Meteor. Soc. Japan, 87A, 1–30, doi:10.2151/jmsj.87A.1.

3) Louf, V., A. Protat, R.A. Warren, S.M. Collis, D.B. Wolff, S. Raunyiar, C. Jakob, and W.A. Petersen, 2019: An Integrated Approach to Weather Radar Calibration and Monitoring Using Ground Clutter and Satellite Comparisons. J. Atmos. Oceanic Technol., 36, 17–39,

https://doi.org/10.1175/JTECH-D-18-0007.1

4) Seto, S. and T. Iguchi, 2015: Intercomparison of Attenuation Correction Methods for the GPM Dual-Frequency Precipitation Radar. J. Atmos. Oceanic Technol., 32, 915–926, https://doi.org/10.1175/JTECH-D-14-00065.1

5) Silberstein, D.S., D.B. Wolff, D.A. Marks, D. Atlas, and J.L. Pippitt, 2008: Ground Clutter as a Monitor of Radar Stability at Kwajalein, RMI. J. Atmos. Oceanic Technol., 25, 2037–2045, https://doi.org/10.1175/2008JTECHA1063.1

---

## Author Comment (AC1) · 8 Nov 2019

**Final response in the interactive discussion**

Dear Referees, dear Editor,

thank you for your critical and constructive comments and suggestions for the improvement to our manuscript "Using ground radar overlaps to verify the retrieval of calibration bias estimates from spaceborne platforms". We would like to use the occasion in order to apologize for the delayed response which was due to personal issues subject to Mrs Crisologo moving from Europe to the US, and to a new job.

In this document, we would like to provide our responses to the comments of each of the three referees in one single document. The referee comments turned out to be very helpful. Based on these comments, we suggest several changes to the manuscript which we will outline in detail on the following pages.

For that purpose, we will show the referee comments in **black** font, and our responses in **blue**. For the sake of clarity, we have deleted some parts of the referee reports which do not contain specific criticism or suggestions. These parts which were not reproduced are marked as [...].

We hope that the suggested changes sufficiently address the referees' concerns, so that we can, given the approval of the editor, finalize the revision of our manuscript.

Sincerely, Irene Crisologo and Maik Heistermann

**Reviewer 1**

[...] the study seems to have overlooked the fact that SR measurements are not perfect either and one could argue also require a similar quality weighting. Yes, the SR reflectivity measurements used in this paper have been corrected for attenuation and this correction is only an estimate. The 2A25 and 2ADPR algorithms rely on the surface-reference technique (Meneghini et al. 2000) and Hitschfeld and Borden method (Hitschfeld and Bordan 1954) to correct the SR measured reflectivity. These techniques can fail or provide poor estimates when multiple scattering and/or non-uniform beam filling may be present, which typically occurs within deep convective precipitation, even at Ku-band (Munchak 2018). As a result, the GR calibration offsets determined by this study may be in error, at least during intense convection. Therefore the authors must address this concern about the quality of the SR measurements, primarily when intense precipitation is included within the matched sample volumes.

We thank the referee for this comment. Obviously, he his right. In Crisologo et al. (2018), we have discussed this issue on page 5233, and also suggested to extend the quality weighting framework to SR reflectivity measurements, particularly with regard to *"the level of path-integrated attenuation (as e.g. indicated by the GPM2AKu variables pathAtten and the associated reliability flag(reliabFlag)) or the prominence of non-uniform beam filling (which could e.g. be estimated based on the variability of GR reflectivity within the SR footprint; see e.g. Han et al., 2018)."* We have not achieved the extension of the framework to the SR measurements, yet. And while we are hesitant to just reproduce the above statement in the present manuscript, we agree that the issue is too important to just implicitly mention it, as we did on p. 20, I. 20, of the original manuscript (*"[...] we need to continue disentangling different sources of uncertainty for both SR and GR observations [...]"*). Thus, we extended the final paragraph of the manuscript as follows:

"[...] Yet, we need to continue disentangling different sources of uncertainty for both SR and GR observations in order to distinguish actual variations in instrument calibration and stability from measurement errors that accumulate along the propagation path, and to better understand the requirements to robustly estimating these properties from limited samples. That also includes to extend the quality-weighting framework to the quality of SR reflectivity measurements, as already outlined in Crisologo et al. (2018), in particular with regard to the combined effects of attenuation at Ku band and nonuniform beam filling which several authors found to cause systematic errors of SR reflectivity measurements in convective situations (see e.g. Munchak 2018, Deo et al. 2018 and Park et al. 2015 for an in-depth discussion) [...]"

Munchak, S. J., 2018: Remote Sensing of Precipitation from Airborne and Spaceborne Radar, In: Islam, T., Y. Hu, A. Kokhannobsky, J. Wang (Eds.): Remote Sensing of Aerosols, Clouds, and Precipitation, p. 267-299, Elsevier, DOI: 10.1016/B978-0-12-810437-8.00013-X. Deo, A., S.J. Munchak, K.J.E. Walsh, 2018: Cross Validation of Rainfall Characteristics Estimated from the TRMM PR, a Combined PR-TMI Algorithm, and a C-POL Ground Radar during the Passage of Tropical Cyclone and Nontropical Cyclone Events over Darwin, Australia, J. Atmospheric and Oceanic Technology, 35(12), 2339-2358, DOI: 10.1175/JTECH-D-18-0065.1

Park, S., S.H. Jung, G. Lee, 2015: Cross Validation of TRMM PR Reflectivity Profiles Using 3D Reflectivity Composite from the Ground-Based Radar Network over the Korean Peninsula, J. Hydromet., 16(2), 668-687, DOI: 10.1175/JHM-D-14-0092.1

Furthermore, we would like to refer to our response to comment #4 of referee #2 who wondered whether the success of the bias estimation and also its interpolation over time might be related to the occurence of convective precipitation (we did not detect a clear relationship in our data).

Another thing that could use additional explanation is the results presented in Table 3. The GR becomes less biased with time, and the relative improvement amongst the three interpolation techniques decreases with time. It even seems that some optimal calibration is attained by 2016. The authors should expand on this and suggest plausible causes for these trends.

We agree that this merits an explicit discussion. Yet, we need to be careful with the terminology: Table 3 (Table 4 in revised manuscript) does not represent the level of miscalibration, but only the inconsistency/mismatch between the two ground radars (in terms of their mean absolute reflectivity difference). While we would expect that a better calibration of each of the two ground radars would result in a better consistency even before any bias correction, the mismatch might also affected be affected by other factors (such as sample size). In fact, with the revised analysis procedure (conversion between Ku- and C-band, exclusion of samples within and above the bright band, see comment #2 of referee #3), the best agreement between SUB and TAG before any bias correction is achieved in 2014 (not in 2016). At the same time, looking at Fig. 5 (both original and revised manuscript), we see that the optimal level of calibration for both radars is rather obtained in 2016 (as SUB still appears to have a pronounced positive bias in 2014). That observation has already been emphasized in the original manuscript, p. 12, II. 29 ff.: "[...] both SUB and TAG are dramatically underestimating at the beginning of operation in 2012, where the underestimation of the TAG radar is even more pronounced. From 2014, the calibration improves for both radars."; and is in line with the referee comment. Unfortunately, we can only speculate that such an improvement has been caused by specific changes in maintenance standards and/or hardware. The lack of maintenance protocols that could corroborate such speculation has briefly been addressed in the conclusions, p. 20, II. 12 ff., of the original manuscript: "[...] maintenance protocols of the affected ground radars would be very helpful in interpreting and interpolating time series of calibration bias estimates. Such records were unavailable for the present study, which made it hard to understand the observed variability of calibration bias estimates."

So, coming back to Table 3 (Table 4 in revised manuscript): it needs to be interpreted with care. Our main conclusion from it is that, on average, the use of interpolated bias estimates is preferable over using uncorrected reflectivities; and that, on average, the moving window approach appears to be a good compromise between linear interpolation and a seasonal average. In order to clarify that, and also to consider the numerical changes in the results after our revised analysis procedure, we rewrote the corresponding paragraph in section 4.4 (p. 17 of the original manuscript) as follows:

"Table 4 provides an annual summary of the mean absolute differences in reflectivity between the two ground radars, without bias correction and with correction of bias obtained from different interpolation techniques. Most importantly, the mean absolute difference between the radars is always lower after correction, irrespective of the year or the interpolation method. Hence, it appears generally preferable to use interpolated calibration bias estimates to correct GR reflectivities, instead of not correcting for bias - even for those periods in which no valid SR overpasses are available. In total, the 30-day moving average slightly outperforms the other two interpolation methods; only in 2016, the seasonal average performs best. The performance of the moving average suggests that it is possible for the calibration of radars to drift slowly in time, with variability stemming from sources which are yet difficult to disentangle. It is also worth mentioning that, for 2016, the mismatch between SUB and TAG before bias correction is guite high (4.3 dB). That is not expected since the calibration of both radars appears to have improved over time (see section 4.2 and Fig. 5). So while the bias correction clearly improves the GR consistency in 2016 (e.g. to a value of 1.6 dB when using a seasonal average for interpolation), we have to suspect that other sources of uncertainty, together with the effect of limited samples sizes, affect the comparison of the two ground radars: e.g. uncertainties in beam propagation, or residual errors in the quantification of path-integrated attenuation and beam blockage."

**[...]**

Minor Comments:

1) Table 1...indicated whether the transmit type of each radar (e.g., SHV or alternating H/V) The type (simultaneous) has been added to Table 1.

2) Section 2.3: NASA, 2017 reference is missing from bibliography NASA, 2017 reference has been added to the bibliography.

3) Section 2.3: Suggest expanding upon the parameters used from TRMM/GPM instead of simply referring to Table 3 of Warren et al. (2018)

Table 2 has been added reflecting the parameters used for analysis has been added in Section 2.3.

4) Figure 3...clarify the vertical reference of the scans (e.g., what elevation angle or constant altitude)

The elevation angle is mentioned in the caption to be 0.5 degrees. We made it more explicit by stating "elevation angle" instead of just "elevation".

5) Section 3.6...7 th paragraph...subscript Qmatch? The subscript has been applied accordingly.

6) Figure 4d...define the dashed line above the histograms.

Thank you for noticing this oversight. The definition is added as:

"The dashed lines represent the distribution of reflectivity differences of all points, when no filter is applied."

7) Figure 7..."samples with significant number of matches"...how many is significant? Thank for pointing out this ambiguity. The figure was made based on GR-GR pairs that had more than 100 matches. This detail has been added to the text.

The figure caption now says:

The differences between the inter-radar consistency before and after correcting for the ground radar calibration biases following a rolling window averaging for GR-GR pairs with more than 100 matches. The hollow (filled) circles represent the daily mean before (after) correction. The line color represents an improvement (green) or a decline (pink) in the consistency between the two ground radars.

[...]

**Reviewer 2**

**[...]**

The following major point needs to be addressed by the authors:

1. The line numbers are preferred to be continuous and clear in the entire manuscript otherwise referencing becomes difficult (see for example page 1, 2 and 3).

Thank you for the comment, we didn't realize the numbering was out of order. The document was created from a template given by the journal. We will bring this issue up to them.

2. Figure 2 quality seems to be compromised which could be improved

**We replaced the figure by a version with higher resolution.**

3. In Section 4.3, line 5 "..the value of mean difference amounts to -4.6 dB" and the authors suggest this deviation possibly due to "..systematic sources.." (line 10). I was wondering if the inherent behaviour of analysing a subset of data from the wider data has been examined here. The overlap data is a subset of the wider SUB/TAG radar data and the bias estimates are obtained from the wider GR-SR data (and not just the overlap data). Maybe highlighting the overlap region data in the SUB-TRMM and TAG-TRMM before and after correction in Figure 6 could help in understanding this deviation, which means adding two more panels similar to (a) and (b) but for GR-SR bias corrected (highlighting the overlap region data). This would show the relative position of the overlap data about the 1:1 line after the bias correction. While this would be an eyeball analysis, a better approach would be to do some statistical analysis for the overlap data after correction. The authors should check this for all the other cases.

We appreciate the referee comment very much. As we understand it, the referee basically wonders whether any bias estimate from the entire ground radar domain can be assumed to be representative for the region of GR overlap - and vice versa. He thus suggests to analyze whether the region of overlap "sticks out" in any kind when we compare GR against SR reflectivity. Yet, we are hesitant to do so. Why? Because it adds another layer of complexity to the analysis, while the expected insight remains, in our opinion, unclear. This is because such an analysis would not allow us to better understand the remaining "systematic sources of error".

The fundamental assumption of this study is that any instrument bias will uniformly affect GR reflectivity across the entire GR domain. Let's picture an ideal case with the absence of any other sources of error except a GR instrument bias. If we now use our bias estimate from the SR-GR comparison to correct both ground radars, the matched GR reflectivities in the region of overlap should perfectly line up along the 1:1 line. Thus, any systematic deviation from the 1:1

line must be due to a systematic source of error which is not uniform across space. Yet highlighting the region of overlap by e.g. an additional figure panel will not help us to pinpoint that source of the error, particularly since the rainfall events are highly heterogeneous in space: if, between all the admittedly terrible scatter, the region of overlap will in fact somehow stick out from the rest of the radar domain, we will still not know whether the unknown source of error is inside or outside that region. Our fundamental motivation (as reflected by the title of our paper) of analysing matched GR reflectivities in the region of overlap is the verification of our quality-weighted matching framework. Obviously and not too surprisingly, it is not perfect yet; and the only way to better understand the systematic errors that are not yet captured, is to experiment with additional quality variables in our quality-weighting framework (potentially expanding to SR quality as well), or to improve the representation of the existing quality variables, in order to see whether we can improve the consistency of SR-GR and GR-GR matches. Hence, we would like to leave Fig. 6 as it is, if the referee agreed.

4. The authors could also examine the nature of the precipitation types studied here (fraction of convective or stratiform) as this could significantly affect the bias correction estimates (underestimation or overestimation: for more detail see studies such as Park et al 2015 and Deo et al 2018). This could help particularly to explain lines 13 -15 (sec. 4.4) ".....17 out of 121 days, .... an increase of more than 1dB in the absolute mean differences" (as given in Figure 7 also).

We appreciate the suggestion. It is also related to a comment of referee #1 who pointed out that SR reflectivity measurements are subject to larger and possibly systematic errors in convective situations, potentially as a result of attenuation correction and non-uniform beam filling. In order to address the suggestion, we investigated the influence of the precipitation type on the performance of bias interpolation over time: for that purpose, we interpolated the percentage of convective matched samples from both SUB and TAG SR overpasses, in the same way we interpolated the bias estimates. That way, we were able to quantify the potential influence of convective samples on the interpolated bias used to correct GR reflectivity. However, as shown in Fig. A of this response letter, we did not find any clear relationship between the occurrence of convection and the consistency of the two ground radars in the region of overlap after bias correction. That does not mean that the presence of convection does not affect the bias estimate from SR overpasses (as found by Deo et al. 2018 and Park et al. 2015). It just means that it does not become apparent in the present verification scheme. Still, a thorough inclusion of SR product data and metadata with regard to SR attenuation still appears to be a promising future extension of the quality-weighting framework. For the present manuscript, however, we decided to not explicitly include this aspect in the analysis, but to highlight the perspective in the final paragraph of the manuscript (as a response to a similar comment of referee #1). In that context, we also refer to the studies of Deo et al. 2018 and Park et al. 2015, as suggested by referee #2.

Fig. A: Like Fig. 7 in the manuscript, this figure shows the differences between the inter-radar consistency before and after correcting for the ground radar calibration biases following a rolling window averaging for samples with significant number of matches. In addition, we show the interpolated fraction of convective samples for SR overpass events with SUB and TAG. The idea behind this approach is to show the influence of bias estimates from SR overpass events with regard to the presence of convection. In other words: if the bias estimation from an SR overpass was degraded by convection, any interpolation of such bias estimate should also degrade inter-radar consistency. However, the figure does not reveal any relationship between the occurrence of pink lines (which indicate that using the interpolated bias degrades inter-radar consistency) and the occurrence of convection.

5. Use of abbreviations such as SUB, TAG, GR, SR, PIA and etc. needs to addressed- use it consistently throughout the manuscript otherwise it distracts a reader: As an example, in lines 13 – 18 (sec. 3.6) there is a combination of SUB, TAG, Subic, Tagaytay in just one paragraph.

Thank you for pointing out these inconsistencies in the manuscript. The abbreviations have been defined at the first instances and replaced accordingly throughout the manuscript.

A list of minor comments/suggestions is as follows:

1. Change "spaceborne" to "space-borne" throughout the manuscript "Spaceborne" has been replaced with "space-borne" throughout the manuscript

Section 1

2. Line 1 "...observations are the key..." remove "the". Same line add article "a" before "large" and "high".

The suggested correction has been applied.

3. Line 3, change "errors" to "error" in "The estimation errors.." The suggested correction has been applied.

4. Two lines before line 5: change "..- let it be.." to "..be it.." The corrected has been applied as suggested.

5. Line 5, introduce PIA abbreviation here and "On top..." is confusing, please rephrase. The PIA abbreviation is defined. "On top" has been replaced with "In addition" to remove confusion.

6. Line 10, remove "...maybe surprising to some..." The line has been removed as suggested.

7. Line 32, change "finally" to "recently" in "...and finally by Warren..." Change has been made as suggested.

8. Page 3, line 16, change "latter" to "last" in "The latter item…" Change has been made as suggested.

9. Rephrase line 24-27 (Page 3). Remove " : " add a "and" before "section 3" in "...data sets: section 3...". Line 26, add full stop after "... "bias estimation" and then "Section 4 will present…results followed by the conclusion in section 5". The paragraph has been rephrased as suggested.

Section 2

10. Line 29 (page 3), Rephrase to "The Philippine weather agency, known as Atmospheric....(PAGASA), maintains..."

The sentences has been rephrased to "The Philippines' weather agency, known as the Philippine Atmospheric... (PAGASA), maintains..."

11. Line 32-33 (page 3-4), replace "at" with "with" and "inhabitants" with "population" in "…area at approximately 13 Million inhabitants."

The sentence has been edited to say "with a population of approximately 13 Million."

12. Figure 1: Make the x and y label fonts consistent.

We patterned the coordinate labeling to the USGS style, we think this is a good way to make the x and y coordinate labels less cluttered with trailing zeros but still readable.

13. Page 4 line 5, define "a.s.l" or use long format "above sea level" in "...532 m a.s.l" The definition for "a.s.l." has been added at the first instance.

14. Page 4, line 8, change "available" to "given". Change made as suggested.

15. Page 4, line 11, "Band" should be lower case in "..C-Band..."

Change made as suggested.

16. Page 4, line 12, change to "...sits on" Change made as suggested.

17. For sec. 2.1. and 2.2, see comment 5 in "Major comments" After defining the SUB and TAG radar names for Subic and Tagaytay, respectively, the abbreviations have been made consistent throughout the manuscript.

18. Page 5, line 11, rephrase "...collected..." "collected" is changed to "obtained"

19. Page 5, line 12, remove "The" from "The data..." Change made as suggested.

20. Page 5, line 12, "...same as specified..." should be "...same as those specified..." Change made as suggested.

Section 3

21. Page 5, Line 3, "(see 2)" should be "(see section 2)" Change made as suggested.

22. Page 5, Line 7, remove bold emphasis of words or sentences which also applies to other sections.

Bold emphasis of phrases have been removed.

23. Page 6, lines 22-23, check the usage of brackets. The extra brackets have been removed.

24. Line 23, remove italicised emphasis of words or sentences which also applies to other sections. Could place them in quotation marks if explicitly required. The italicization of "imperative" has been removed.

25. Section 3.4, page 7, line 7, "dual-pol"? "dual-pol" has been explicitly stated as "dual-polarization"

26. Page 8, line 13, delete repeated "copolar cross-correlation" Change made as suggested.

27. Page 8, line 14, delete "and" in "...and differential propagation phase ..." Change made as suggested.

28. Page 8, line 18, define KDP. KDP has been defined.

29. Page 8, line 30, define DEM. DEM has been defined.

30. Page 8, last sentence, add "a" before "total" in "... corresponds to total.." and also before "complete" at the end of the sentence Change made as suggested.

31. Figure 3, I believe "...corresponding elevation angle." should have been "...corresponding sweep angle.)

The term "elevation angle" refers to the sweep angle. For consistency, as we have been using "elevation angle" throughout the manuscript, we opt to keep "elevation angle" in this caption.

32. Page 9, line 17, add "a" before "…very high beam…". Change made as suggested.

33. Page 9, line 17, state which "…higher elevations.." Is it > 0.5 or 1.5? Sorry for the confusion. In this part, we refer to higher elevations as those > 0.5 degrees. This has been clarified in the manuscript:

"higher elevation angles (>0.5°)"

**34. Page 10, Equation 5, define Kr,s**

Thank you for pointing out this oversight. The term "Kr,s" is ambiguous and has been replaced instead with Ai, which refers to a one-way path-integrated attenuation at the ith radar bin. Correspondingly, Kmax and Kmin have been replaced with Amax and Amin to signify the minimum and maximum attenuation thresholds.

**Section 4 and 5**

35. Page 11, line 28, define how much is sufficient in "...sufficient radar bins". The sentence has been updated to "where there are more than 900 radar bins with precipitation in the region of overlap".

36. Page 11, line 28, is the time LT or UTC?

Thank you for pointing out this ambiguity. The times are in local times, we clarified this in the text.

"The scan times are 06:55:14 and 06:57:58 (local times) for the SUB and TAG radars, respectively."

37. Page 12, line 9, rephrase to "...underestimation by the TAG..." Change made as suggested

38. Page 12, line 15.Rephrase "Remembering item (2)..." to " Considering component (2)..." Change made as suggested

39. Page 12, Line 19, remove brackets in "... (SR-GR or GR-GR)... Change made as suggested

40. Page 12, line 27, delete "is" from "The first panel is corresponds…" Change made as suggested.

41. Page 12, line 28, "Section 3.2" instead of "Section III.2" Change made as suggested.

42. Page 12, line 33, change "drastic" to "severe" Change made as suggested.

43. Figure 4, panel d, what do the dashed lines represent? Also add article "a" before "low" and "high"

Thank you for noticing this oversight. The definition is added as:

"The dashed lines represent the distribution of reflectivity differences of all points, when no filter is applied."

44. Page 14, line 6 define "sufficient number". Also add "In" before "That" in "That way, we…" The "sufficient number" has been explicitly defined as "as at least 30 matched GR samples". The succeeding sentence has also been updated to "In that way,[...]"

45. Page 14, line 8, change "item (2)" to "component (2)". Also replace "...in which..." with "where".

Changes made as suggested.

46. Page 14, line 10, replace "...took place right..." with "occurred" Change made as suggested.

47. Page 14, line 11, rephrase "according to Figure 6" to "see Figure 6" and put in brackets. Also add "during" before "...the so called Habagat...". Changes made as suggested.

48. Table 2, define Npts. Npts have been spelled out as "number of points". 49. Page 15, line 7, replace "massive" with large. This also applies to other sections. "Massive" has been replaced with "Large".

50. Page 15, line 20, rephrase to "The question now is..." Change made as suggested.

51. Page 15, line 25, add "a" before "…few examples". Change made as suggested.

52. Figure 6, is time LT or UTC? Also include a colour bar to show the scale density and add "respectively for" after "consistencies" in (c) and (d)

The time in Figure 6 is local time, we have now indicated this in the manuscript. The color scale for these images are based on the kernel density divided by the maximum number for each scatter plot, such that the darkest colored point for each subplot corresponds to a value of 1. This eliminates the need to add a color bar.

53. Page 18, lines 11-12, rephrase.

The sentences have been rephrased to read:

"The length of the bar shows the magnitude of the change, while the color of the bar signifies improvement or degradation of consistency between the ground radars. A green bar denotes that the absolute value of the mean difference after correction has decreased, \i.e. the mean difference after correction (filled circles) is closer to zero than before correction (unfilled circles). A pink bar denotes an increase in the absolute value of mean difference between the two radars after correction."

54. Page 18, line 16, I believe you meant "unacceptable" rather than "acceptable". We did mean acceptable. The sentence has been rephrased to make this clearer: *"However, we are also able to identify several days for which the bias correction did decrease the absolute mean differences, yet still not to a level that could be considered as acceptable for quantitative precipitation estimation."*

55. Page 15, line 19, check referencing format i.e. for Schwaller and Morris. The referencing has been updated.

56. Page 19, line 31, could use "...main finding" instead of "...main lesson" Change made as suggested.

57. Page 19, line 35, use full form in "single-pol" Change made as suggested.

58. Page 19, line 36, may be use numerals (i,ii and iii) to list the approaches Change made as suggested.

59. Page 19, last line, replace "any" with "all" in "On average, any.." Change made as suggested.

60. Page 20, line 8, delete one of the "also". The first "also" has been deleted.

61. Page 20, line 10, replace "Altogether" with "Hence". Change made as suggested.

62. Page 20, line 13, replace "hard" with "difficult" Change made as suggested.

63. Page 20, lines 16-17 please rephrase the sentence after the website link as it is confusing. Thank you for the suggestion, we have edited the sentence as follows:

*"With the software code and sample data of our study being openly available (https://github.com/IreneCrisologo/inter-radar), such institutions are now enabled to carry out the analysis presented in this study by themselves, while being able to benefit from cross-referencing the results with internal maintenance protocols."*

64. Page 20, line 18, "..in periods" should be ("...during periods..." Change made as suggested.

65. Page 20, line 22, replace "...estimating..." with "estimate" Change made as suggested.

**Reviewer 3**

**[...]**

**Specific comments**

1) In section 2, descriptions of used data are overall lacked. How to originally calibrate the ground radars? Don't some references and/or descriptions of the ground radars exist? How to treat/correct the precipitation attenuation for ground/spaceborne radars? Does the current study compare what parameters of radar reflectivity with or without the attenuation? This study mentions Crisologo et al. (2018) and Warren et al. (2018), but it makes readers to feel unkindly. Especially, data of the C-band ground radar because this study newly utilizes the data. Please

describe appropriately. I suggest some references of attenuation-correction methods for spaceborne radar data as follows: Iguchi et al. (2009) and Seto and Iguchi (2015).

We regret that some of the information required by the referee appears to be lacking in the manuscript. Yet, we would prefer not to reproduce the description of the S-band radar data provided in Crisologo et al. (2018) and of the SR (TRMM/GPM) radar data as provided in Warren et al. (2018). While we understand that it is more convenient for the reader, such a repetition would not be in line with the required conciseness and brevity, as we see it. As this appears to be a matter of style, we would be willing to add corresponding changes if the editor insisted. With regard to the Tagaytay C-band radar, we describe the data in section 2.2 at a level we consider sufficient. Unfortunately, we are unable to provide details on how the radar were originally calibrated by the operator, as is pointed out in the conclusions section of the manuscript. There exists no official document that elaborates on the ground radar network, just a personal communication with PAGASA personnel who stated that receiver calibration is carried out using an internal test signal.

However, we revised section 2.3 of the manuscript by including the references on attenuation correction for the space-borne radars, as suggested by the referee. It now reads as follows:

"Spaceborne radar data were collected from TRMM 2A23 and 2A25 version 7 (NASA, 2017) for overpass events in 2012-2014, and GPM 2AKu version 5A products (Iguchi et al., 2010) from 2014-2016, during the rainy season of June to December. The products include, among others, an attenuation correction of observed reflectivity (see e.g. Iguchi et al., 2009, for the TRMM precipitation radar, and Seto et al., 2015, for GPM) [...]"

Iguchi, T., T. Kozu, J. Kwiatkowski, R. Meneghini, J. Awaka, and K. Okamoto, 2009: Uncertainties in the Rain Profiling Algorithm for the TRMM Precipitation Radar. J. Meteor. Soc. Japan, 87A, 1– 30, doi:10.2151/jmsj.87A.1.

Seto, S. and T. Iguchi, 2015: Intercomparison of Attenuation Correction Methods for the GPM Dual-Frequency Precipitation Radar. J. Atmos. Oceanic Technol., 32, 915–926, https://doi.org/10.1175/JTECH-D-14-00065.1 5)

If we understood the referee correctly, he also inquired whether the C-band radar reflectivity was corrected attenuation (for the intercomparison). Here, we would like to refer to p. 3, II. 6 ff., of the original manuscript: "[...] Instead of attempting to correct GR reflectivities for PIA, we explicitly acknowledge the uncertainty of any PIA estimate by assigning a low weight to any GR bins that are substantially affected by PIA [...]". In order to emphasize the fact that we did not correct for attenuation, but rather filtered measurements affected by attenuation, we modified the second paragraph of section 3.4 as follows:

**"[...] In this study, we did not correct the ground radar reflectivity for attenuation. Instead, we require PIA estimates as a quality variable to assign different weights of GR reflectivity samples when computing quality-weighted averages of reflectivity (see section 3.6) [...]"**

2) This study uses the two ground radars at frequencies of S- and C-bands and the spaceborne radars at a frequency of Ku-band. How to consider the difference in a frequency? From Crisologo et al. (2018), a conversion due to the Mie-scattering effect from Ku-band to S-band is empirically considered in this study. However, the conversion among S-, C-, and Ku-bands is not described anything. Is the residual difference of the two ground radars mixed with the frequency difference? Please describe explicitly.

We apologize for not having addressed the issue of different radar frequencies in the manuscript. In fact, we had used, in our original analysis, the same conversion function for Ku-band to C-band reflectivity as for the S-band radar. We thus thank the referee for pointing out that neglect, and revised our analysis accordingly: Before estimating the GR bias (Tagaytay radar) from SR overpasses, we convert the SR reflectivity from Ku- to C-band using an empirical function used by Louf et al. (2019), Eq. 5, which is based on T-matrix scattering simulations. Since Louf et al. state that the function was only valid for liquid precipitation, we excluded all matching samples from within and above the bright band (using the bright band detection from the SR data). For the sake of consistency, we followed the same approach (i.e. discarding samples from within and above the bright band) for the S-band radar (Subic). Apart from recomputing the results with these changes, we modified the corresponding paragraph in section 3.2 (SR-GR matching) of the revised manuscript in order to clarify the procedure:

In section 3.2: "[...] In this study, we extend it to the TAG radar. Since the two radars are operating under the same scanning strategy and spatial resolution, the thresholds applied in filtering the data are kept the same as in the SR-SUB comparison described in Section 3.2, Table 2, of Crisologo et al. (2018), except that we considered samples only from below the bright band (as specified by the bright band detection in the SR product). That methodological adjustment was necessary due to the conversion between Ku and C-band reflectivity, which accounts for the systematic effect of different measurement frequencies: For that conversion, we used an empirical function published by Louf et al. (2019), Eq. 5, which was derived from T-matrix scattering simulations. According to the authors, that function is only valid for liquid rain; hence we excluded samples from within and above the bright band. The same was done for the S-band radar, in order to keep the matching procedure consistent between SUB and TAG. The conversion from Ku- to S-band reflectivity was implemented using the functions published by Cao et al. (2013). Further details of the SR data specifications and the matching procedure can be found in Crisologo et al. (2018)."

We furthermore revised all the text and figures in the manuscript in order to account for the new results (although the overall conclusions are not affected by these changes).

Please note, however, that we decided not to convert between S- and C-band frequencies for the GR-GR matching procedure. We modified section 3.3 in order to explain the reasons behind that decision:

In section 3.3: "We compare the reflectivities of both ground radars in the overlapping region to quantify the mean and the standard deviation of their differences, and thus the effectiveness of the quality-weighting and the relative calibration procedure. Please note that we do not explicitly account for differences in the reflectivity factor between S-band and C-band due to resonance effects, although Baldini et al. (2012) found that for very high reflectivities and very high median volume diameters of the drop size distribution, the deviation between the reflectivity factors of S-band and C-band can reach up to a maximum of 3 dB. Yet, we assume that, in such a scenario, the uncertainty introduced by path-integrated attenuation and its correction for C-band is more important, and at the same time implicitly addressed by the quality-weighting framework. In order to compare reflectivities from different radars [...]"

**Baldini, L., V. Chandrasekar, and D. Moisseev, 2012: Microwave radar signatures of precipitation from S band to Ka band: application to GPM mission, European Journal of Remote Sensing, 45(1), 75-88, DOI: 10.5721/EuJRS20124508.**

3) This study tries a temporal adjustment of calibration changes for ground radars from only infrequent matchups with spaceborne radars. The relative calibration adjustment with ground clutter (e.g., Silberstein et al. 2008, Louf e t al. 2019) is one of typical methods. Each of the relative calibration methods with the ground clutter and the matchup with spaceborne radar has its merits and demerits. Please discuss in the manuscript.

We appreciate that comment very much because relative calibration methods that rely on ground clutter as a reference could very well complement the calibration approach using space-borne radars: first, they could help to verify the temporal variability of bias estimates obtained from SR overpasses; second, they could support the interpolation of bias estimates in time, e.g. as a covariate. Yet, as the referee is most likely well aware, Louf et al. (2019) did exactly this, and we thank the referee for this reference which we had not been aware of, yet. But instead of going into depth discussing pros and cons of both approaches, we'd like to just briefly add this as a perspective enhancement to the conclusions section, p. 20 (as follows), including the reference to Louf et al.:

"[...] Altogether, it still appears difficult to interpolate such a volatile behaviour, even if we considered the actual calibration bias estimates from the SR overpasses as quite reliable. A way to further investigate that behaviour would be to complement the analysis by relative calibration techniques that use ground clutter returns as a reference (e.g. Silberstein et al. 2008). Although such techniques only allows to detect changes in calibration relative to a baseline, they can be applied to each volume cycle and thus inform us about dynamics at a high temporal resolution and coverage. That way, we could support the interpolation of bias estimates obtained from SR overpasses, or scrutinize the temporal variability of such estimates.

An application and in-depth discussion of this concept has just recently been provided by Louf et al. (2019) with the example of the C-band weather radar in Darwin, Australia."

Louf, V., A. Protat, R.A. Warren, S. M. Collis, D.B. Wolff, S. Raunyiar, C. Jakob, W.A. Petersen, 2019: An Integrated Approach to Weather Radar Calibration and Monitoring Using Ground Clutter and Satellite Comparisons, J. Ocean. Atm. Techn., DOI: 0.1175/JTECH-D-18-0007.1

Silberstein, David S., David B. Wolff, David A. Marks, David Atlas, and Jason L. Pippitt. "Ground Clutter as a Monitor of Radar Stability at Kwajalein, RMI." *Journal of Atmospheric and Oceanic Technology* 25, no. 11 (November 1, 2008): 2037–45. https://doi.org/10.1175/2008JTECHA1063.1.

**Technical corrections**

1) Page 2 line 23: The sentence in a parenthesis after "estimation errors(in..." is too long, so I might be better to replace it. For examples: "The estimation errors are defined as retrieval errors of the precipitation rate... the radar reflectivity factor Z; then the errors are caused by...".

We changed the original sentence (on p. 1, II. 23 ff.) as follows:

"We define estimation errors as errors that occur in the retrieval of the precipitation rate R from the radar's prime observational target variable, the radar reflectivity factor Z. These errors are caused mainly by the unknown microphysical properties of the target - be it meteorological or non-meteorological."

2) Page 2 line 4: I do not understand "let it be ...". What does it mean?

Please see our changes to the same sentence as a response to the previous comment (1)

3) Please specifically indicate the frequency or wavelength of the radars in Sections 2.1 and 2.2.

4) "Bandwidth" in Table 1 is wrong. I think "Frequency" is appropriate.

"Bandwidth" has been replaced with "Frequency"

5) Page 7 line 27 and Page 15 first paragraph in Section 4.4: SR has been already defined at Page 2. Why did you redefine SR? In Section 5 (summary), I understood the redefinition as a refresh.

Thank you for pointing that out. The redefinitions of SR in Page 7 and Page 15 have been removed.

The corresponding sentences now reads:

[...] calibration approach by using the space-borne-radar (SR) as a reference.

The space-born radar SR platform rarely overpasses both GR radar domains [...]

6) Page 7 line 33: I can not find Table 3 in Crisologo et al. (2018). Is it Table 2 in Crisolog et al. (2018)? Please indicate the correct number.

Apologies for this oversight, the table number has been corrected to Table 2.

7) Table 2: -5 and -7 is should be -5.0 and -7.0 if the significant digit of those values is correct in this study.

The numbers have been changed to -5.0 and -7.0.

8) Some references lack information such as URL (e.g. Iguchi et al. 2010, Jone et al. 2014). Please check

https://www.atmospheric-measurementtechniques.net/for\_authors/manuscript\_preparation.html. Incidentally, Iguchi et al. (2010) is too old for a reference. Please update appropriately as follows:

https://pmm.nasa.gov/resources/documents/gpmdpr-level-2-algorithm-theoretical-basisdocumen t-atbd.

The GPM reference has been updated to Iguchi et al. 2018. The SciPy reference has been updated from Jones et al. (2014) to Virtanen et al. (2019).

Iguchi, Toshio, Shinta Seto, Robert Meneghini, Naofumi Yoshida, Jun Awaka, Minda Le, V Chandrasekar, Stacy Brodzik, and Takuji Kubota. "GPM/DPR Level-2 Algorithm Theoretical Basis Document," 2018.

https://pmm.nasa.gov/resources/documents/gpmdpr-level-2-algorithm-theoretical-basis-docume nt-atbd.

Virtanen, Pauli, Ralf Gommers, Travis E. Oliphant, Matt Haberland, Tyler Reddy, David Cournapeau, Evgeni Burovski, et al. "SciPy 1.0-Fundamental Algorithms for Scientific Computing in Python." CoRR abs/1907.10121 (2019). http://arxiv.org/abs/1907.10121.